



# Remote sensing data processing by multivariate regression analysis method for iron mineral resource potential mapping: A case study in Sarvian area, central Iran.

Edris Mansouri[1], Faranak Feizi[2*], Alireza Jafari Rad[3], Mehran Arian[4]

1- Department of Geology, Science and Research Branch, Islamic Azad University, Tehran, Iran

2- Department of Mining Engineering, South Tehran Branch, Islamic Azad University, Tehran, Iran

3- Department of Geology, Science and Research branch, Islamic Azad University, Tehran, Iran

4- Department of Geology, Science and Research Branch, Islamic Azad University, Tehran, Iran

*corresponding author

Fax number: +98 021 88830012; Cell: +98 912 3006753; faranakfeizi@gmail.com.

## ABSTRACT

This paper used multivariate regression to create a mathematical model (with reasonable accuracy) for iron skarn exploration in the region of the interest and generalizing multivariate regression in Mineral Prospectivity Mapping (MPM) field. The main target of this manuscript is to exert multivariate regression analysis (as a MPM method) to iron outcrops mapping from northeast part of the study area to discover new iron deposits in other parts. Two types of multivariate regression models as two linear equations were employed to discover new mineral deposits. The Aster satellite image bands (14 bands) sets as Unique Independent Variables (UIVs) and iron outcrops map as dependent variables were used for MPM. According to the results of p-value, $R^2$ and $R_{adj}^2$, the second regression model (which was a multiples and exponents of UIVs) was the fitted model versus other models. Also the accuracy of the model was confirmed by iron outcrops map and geological observations. Based on field observation iron mineralization occurs as contact of limestone and intrusive rocks (skarn type). Iron minerals consist dominantly of magnetite, hematite and goethite.

**Key words:** Multivariate regression, Mineral Prospectivity Mapping, Iron, Sarvian



## 1. INTRODUCTION


Diagnosing futuristic zones and finding new mineral deposits in the region of interest, is the
definitive main of mineral investigation. One way to achieve this aim is Using satellite image
processing for identify Mineral Prospectivity Mapping (MPM) (Carranza, 2008; Abedi et al., 2013;
Golshadi et al., 2016 and Feizi et al., 2012).
The utilization of satellite images for mineral investigation has been extremely effective in
indicating out the attendance of minerals. Likewise, remote sensing gives the synoptic view, which
is useful in distinguishing proof and delineation of different land frames, linear features, and
structural elements (Feizi and Mansouri, 2013b).
The main objective of this manuscript is to use multivariate regression analysis (as a MPM
method) to pixel values of Aster satellite image from north-east part of the study area to identify
new iron deposits in other parts. Two types of multivariate regression models utilized to find new
mineral deposits, using pixel values of Aster satellite image bands (14 band) sets as Unique
Independent Variables (UIVs) and Iron outcrops surface (digitized by geology map of study area
(scale 1:5000) and check field) data as dependent variables.
Regression analyses have been utilized as a part of numerous logical fields, such as
geosciences.
Identification of stream sediment anomalies have been used by multiple regression analyses
(e.g., Carranza, 2010a; Carranza, 2010b). Likewise, multivariate regression has been effectively
utilized by Granian et al. (2015) to display subsurface mineralization from lithogeochemical
information. Granian et al. (2015) utilized four types of multivariate regression models to depict
significant surface geochemical anomalies for acknowledgment subsurface gold mineralization
utilizing borehole data as dependent variables and surface lithogeochemical data as independent
variables.
This paper utilized multivariate regression to make a mathematical model (with sensible
precision) for iron potential zones investigation in the region of the interest and summing up
multivariate regression in remote sensing field.

## 2. STUDY AREA


The Sarvian area is located in the Orumieh-Dokhtar magmatic arc in Central of Iran (Fig. 1a).
This magmatic arc is the most imperative metallogenic area inside the district and hosts the
majority of the larger metals deposits such as lead, zinc, copper and iron (Feizi et al., 2016 and
Feizi et al., 2017).
The explored zone determined by Eocene intrusive rocks and carbonates of Qom formation.
Several types of metal and non-metal mineral ore deposits have as of now been reported in the





study area. According to the 1:100,000 geological map of Kahak, the lithology of this part includes
cream limestone with intercalations of marls (Qom formation), dark green, andesitic-basaltic lava,
volcanic breccia, hyaloclastic limestone, green megaporphyritic andesitic-basaltic lava,
rhyodacitic domes, tonalite-quartzdiorite, microquartzdiorite-microquartzmonzo-diorite, granite-
granodiorite, alteration of light green, grey tuff, tuffaceous sandstone and shale with intercalation
of nummulitic sandy limestone and andesitic lava, grey limestone, orbitolina bearing, thick bedded
to massive (Aptian–Albian) (Feizi et al., 2016) (Fig. 1b).
In view of the current confirmations and furthermore contact of intrusive bodies with carbonate
rocks (Qom arrangement) and Iron outcrops  in the north-east of study area, Calcic iron skarn ore
(Sarvian mine) is located in the northeast of study area (Feizi et al., 2017) (Fig. 2).

**3. MULTIVARIATE REGRESSION**
Regression analyses is a good statistical manner for analysing relationships among variables
(Granian et al., 2015). This strategy can show the conduct of an event (a dependent variable) in
light of related variables (some independent variables). In regression analyses, if a dependent
variable called $(Y)$ and independent variables called $(x_i)$, the equation is:
$$Y = f(x_i). \qquad (1)$$

Y could be linear or non-linear function. Linear regression is used for modeling mineral
prospetivity in the Sarvian area. For linear regression Y is defined as follows:
$$Y = a_0 + a_1 x_1 + a_2 x_2 + \cdots + a_i x_i + \varepsilon , \quad i = 1,2,\dots,n. \quad (2)$$


For this function, the constant factor is $a_0$ , the random error is $\varepsilon$ , and the regression
coefficients are $a_i$. If there were $n$ samples in a data set, for each sample $t$ variables were
measured. Thus, function (2) can be as follows:

$$Y_i = \hat{a}_0 + \hat{a}_1 X_{i1} + \hat{a}_2 X_{i2} + \cdots + \hat{a}_t X_{it} + \varepsilon_i i = 1,2,\dots,n. \qquad (3)$$



Equation (3) can be re-written as a matrix. The linear function matrix is:
$$[Y] = [X][A] + [\varepsilon]. \qquad (4)$$

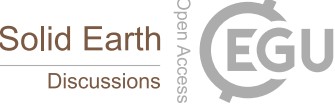


$$[Y] = \begin{bmatrix} Y_1 \\ Y_2 \\ \cdot \\ \vdots \\ Y_n \end{bmatrix}; \quad [A] = \begin{bmatrix} \hat{a}_0 \\ \hat{a}_1 \\ \cdot \\ \vdots \\ \hat{a}_t \end{bmatrix}; \quad [X] = \begin{bmatrix} 1 & X_{11} X_{12} \dots X_{1t} \\ 1 & X_{21} X_{22} \dots X_{2t} \\ & \cdot \\ & \vdots \\ 1 & X_{n1} X_{n2} \dots X_{nt} \end{bmatrix}; \quad [\varepsilon] = \begin{bmatrix} \varepsilon_1 \\ \epsilon_2 \\ \cdot \\ \vdots \\ \epsilon_n \end{bmatrix}. \quad (5)$$


The least squares technique is used for estimating $[A]$ as the coefficient matrix, as follows:

$$[A] = [\textstyle\sum \ ]^{-1}[C] = ([X]'[X])^{-1}[X]'[Y]. \quad (6)$$

The inverse of variance-covariance samples matrix is $[\sum \ ]^{-1}$ and the covariance matrix among
independent variable and samples is $[C]$. Thus by equation 6, the regression coefficients model is
calculated.
In regression analysis, some criteria are required to review. These criteria are as follows:
1. The variance and the mean of the random error should be a constant value and zero,
respectively.
2. The coefficient of determination value ($R^2$) should be examined. This value is calculated as
follows (Granian et al., 2015):

$$R^2 = \frac{\sum_{i=1}^{n}(\hat{Y}_i - \bar{Y})^2}{\sum_{i=1}^{n}(Y_i - \bar{Y})^2} = 1 - \frac{\sum_{i=1}^{n}(Y_i - \hat{Y}_i)^2}{\sum_{i=1}^{n}(Y_i - \bar{Y})^2}. \quad (7)$$

The mean of the variable ($\bar{Y}$), value of $i$th sample ($Y_i$) and estimated value of the $i$th sample
($\hat{Y}_i$) for dependent variables were used in equation 7. The calculated $R^2$ value determined within
[0, 1] range. The value of $R^2$ is close to 1 for well fitted models.
1.  According to the fact that adding independent variables to the model will increasing $R^2$
114       value, the adjusted determination coefficient ($R^2_{adj}$) is defined as follows (Granian et al.,
115       2015):

$$R^2_{adjusted} = 1 - \frac{n-1}{n-t}(1 - R^2). \quad (8)$$



As it was mentioned, $n$ is number of samples (or data) and $t$ is the number of variables (or
regression coefficients). If a set of explanatory variables are introduced into a regression one at a
time, with the $R^2_{adj}$ computed each time, the level at which $R^2_{adj}$ reaches a maximum, and decreases
afterward, would be a well fitted model.
2.    In regression analyses, the model should be fitted to the data. Accordingly, the p-value of
122         the regression model in Analysis of variance (ANOVA) test should be acceptable (less
123         than or equal to 0.05). Also calculating p-value of final coefficients for each model, could
124         help on optimizing and improving the model. This criterion could be considered after
125         choosing the best model.

4. **GEO-DATA SETS PREPARATION**
The iron ore skarn type located in the northeastern of Sarvian area. There are several iron vein
and outcrop in this area. According to the regional geological conditions of the area, the data set
of this iron mine is a good model for exploring the surrounding area. In this paper, satellite imagery
and map the geology of the mine is used as a training area. In the training area, this method can
model the iron outcrops (a dependent variable) based on Aster satellite image bands (some
independent variables) (Fig. 3).
**Figure 3 is about here.**
**4.1. REMOTE SENSING DATA (INDEPENDEBT VARIABLES)**
The ASTER sensor was propelled in December 1999 on board the Earth Observation System
(EOS) US Terra satellite to record sun powered radiation in 14 spectral bands (Table 1). ASTER
provides high-resolution images of the land surface, water, ice, and clouds using three separate
sensor subsystems covering 14 multi-spectral bands from visible to thermal infrared. The
significant resolution scales are 15m, 30m, and 90m in the visible, short-wave IR, and thermal fR,
respectively. ASTER consists of three different subsystems; the Visible and Near Infrared (VNIR),
the Shortwave Infrared (SWIR), and the Thermal Infrared (TIR). To find out more about each
module click on the item of interest (Feizi and Mansouri, 2013b and Mansouri and Feizi, 2016).
Several factors influence the signal measured at the sensor, for example, float of the sensor
radiometric calibration, atmospheric and topographical effects. In this way, Aster data collection
was utilized and radiance correlation, such as wavelength, dark subtract and log residual by
ENVI5.1 software which is basic for multispectral images, were utilized (Mansouri et al., 2015).
In this study after the corrections, pixles size of SWIR and TIR bands based on VNIR3 band
(Panchromatic band) convert to 15 meter, than use layer stacking function to build a new multiband
file from georeferenced images of various pixel sizes, extents, and projections.
**4.2. MAPPING OF IRON OUPCROPS (DEPENDENT VARIABLE)**





The iron ore skarn type located in the northeastern of Sarvian area. There are several iron vein
and outcrop in this area. In order to mapping of iron outcrops in the training area used from
Geological map (1:1000 scale) of iron ore deposit and check field. For preparing of this layer, the
shape file layer of iron outcrops convert to raster file with pixel size of 15 meter.

**5. REGRESSION ANALYSES IN THE STUDY AREA**

Regression analyses needs making proper models. Utilizing multiple, factorial, polynomial and
reaction surface regressions have been utilized as a part of numerous logical fields such as
geosciences (e.g. Granian et al., 2015). Thus, in this study; Model 1 ($Y_1$) was generated as a
multiple linear regression model and Model 2 ($Y_2$) was created from $Y_1$ plus multiplied UIVs. The
formulas of two mentioned models are presented in Table 2. So in summary, two linear equations
($Y_1$ and $Y_2$) were utilized to discover new mineral deposits, using pixel values of Aster satellite
image as independent variables and map of iron outcrops as dependent variables. The models
which were proposed in this paper, had become more complexes respectively 1 to 2, model 2 has
106 coefficients (14 for UIVs, 1 as constant, 91 for multiples of UIVs) and model 1 has 15
coefficients (14 for UIVs, 1 as constant, 0 for multiples and exponents of UIVs) (Table 2).
For assessing the models which are exhibited in Table 2, regression analyses were performed
and the critical criteria which are mentioned before, were examined. The values of the $R^2$ , $R^2_{adj}$
and p-value of ANOVA test of 2 multivariate regression models are provided in Table 3.

Also, Table 4 is presented the calculated coefficients of independent variables in regression
models. Other independent variables which are not mentioned in Table 4 were excluded variables.
The excluded variables have no effect on the models. This means that, excluded variables didn't
have any effect on iron mineralization and behavior of iron outcrop map.

6. **DISCUSSION**
For distinguishing the best model among 2 models, a few criteria are required to review.
Firstly, the variance and the mean of the random error were acceptable for all of regression
models. Secondly, based on Table 4, the p-values of ANOVA test of 2 multivariate regression
models were equal to 0. For regression models the acceptable p-value should be less than or equal
to 0.05. Thus, this criterion confirmed the validity of models without specifying the most
appropriate model.
On the other hand, the value of $R^2$ is close to 1 for well fitted models. The $R^2$ values of
regression models are presented in Table 3. The lowest $R^2$ belongs to $Y_1$ and the highest belongs to
$Y_2$. Thus, $Y_2$ model is better from $Y_1$ model.



According to the fact that adding independent variables to the model will increasing $R^2$ value,
the $R^2_{adj}$ value should be checked. The $R^2_{adj}$ values of regression models are presented in Table 3.
As it was mentioned before, if a set of variables are introduced into a regression, with the
$R^2_{adj}$ computed each time, the level at which $R^2_{adj}$ reaches a maximum, and decreases afterward,
would be a well fitted model. So, according to Table 3, $Y_2$ would be the fitted model versus other
models. Thus, $Y_4$ regression model is the most appropriate model for Mineral Prospectivity
Mapping.
Thus according to the results of p-value (ANOVA test), $R^2$ and $R^2_{adj}$, the Second regression
model ($Y_2$) would be the fitted model versus other models. For generating the mineral prospectivity
map the formula of $Y_2$ was performed in ArcGIS software by raster calculator tool. The normalized
mineral prospectivity map of the study area is presented in Fig. 4.
To assess the exactness of the selected model, the created prospectivity map was checked by
the iron outcrops map in the northeast part of the study area (Fig. 5). The locations of iron outcrops
have appropriate adoption with favorable areas of mineral prospectivity map. In addition the
adaption of prospectivity map with the iron outcrops in the northeast part of the study area, three
target areas with very high favourability, were checked and the prospectivity map was confirmed
by geological observations (Fig. 6). Based on field observation iron mineralization occurs as
contact of limestone and intrusive rocks (skarn type). Iron mineralizations consist dominantly of
magnetite, hematite and goethite. Therefore, the accuracy of mineral prospectivity map confirmed
in the Sarvian area.

## 7. CONCLUSION

The conclusions of this manuscript are presented in summary as follows.
1) The application of multivariate regression analysis (as a MPM method) was confirmed in
the Sarvian area. This paper used multivariate regression to create a mathematical model (with
reasonable accuracy) for iron mineral exploration in the region of the interest and generalizing
multivariate regression in MPM field.
2) Two types of multivariate regression models as two linear equations were employed to
discover new mineral deposits. According to the results of p-value, $R^2$ and $R^2_{adj}$, the second
regression model was the fitted model versus other models.
3) Also the accuracy of the model was confirmed by iron outcrops map and geological
observations. Based on field observation iron mineralization occurs contact of limestone and
intrusive rocks (skarn type). Iron mineralizations consist dominantly of magnetite, hematite and
goethite.



ACKNOWLEDGEMENTS
The authors would like to thank Amirabbas KarbalaeiRamezanali for his helpful suggestions.


**REFRENCES**
Carranza, E.J.M., 2008, Geochemical anomaly and mineral prospectivity mapping in GIS,
Handbook of Exploration Environmental Geochemistry. Elsevier, Amsterdam, 368 p.
Carranza, E.J.M., 2010a, Catchment basin modelling of stream sediment anomalies revisited:
incorporation of EDA and fractal analysis. Geochemistry: Exploration, Environment, Analysis,
10, 365–381.
Carranza, E.J.M., 2010b, Mapping of anomalies in continuous and discrete fields of stream
sediment geochemical landscapes. Geochemistry: Exploration, Environment, Analysis, 10, 171–
232    187.

Feizi, F. and Mansouri, E., 2012, Identification of Alteration Zones with Using ASTER Data
in A Part of Qom Province, Central Iran. Journal of Basic and Applied Scientific Research, 2, 73–
235    84.

Feizi, F. and Mansouri, E., 2013a, Separation of Alteration Zones on ASTER Data and
Integration with Drainage Geochemical Maps in Soltanieh, Northern Iran. Open Journal of
Geology, 3, 134–142.
Feizi, F. and Mansouri, E., 2013b, Introducing the Iron Potential Zones Using Remote Sensing
Studies in South of Qom Province, Iran. Open Journal of Geology, 3, 278–286.
Feizi, F., Mansouri, E. and KarbalaeiRamezanali, A., 2016, Prospecting of Au by Remote
Sensing and Geochemical Data Processing Using Fractal Modelling in Shishe-Botagh, Area (NW
Iran). Journal of the Indian Society of Remote Sensing, 44, 539–552.
Feizi, F., KarbalaeiRamezanali, A. and Mansouri, E., 2017, Calcic iron skarn prospectivity
mapping based on fuzzy AHP method, a case Study in Varan area, Markazi province,
Iran. Geosciences Journal, 21, 123–136.
Granian, H., Tabatabaei, S. H., Asadi, H. H. and Carranza, E. J. M., 2015, Multivariate
regression analysis of lithogeochemical data to model subsurface mineralization: a case study from
the Sari Gunay epithermal gold deposit, NW Iran. Journal of Geochemical Exploration, 148, 249–
250    258.



Golshadi, Z., KarbalaeiRamezanali, A. and Kafaei, K., 2016, Interpretation of magnetic data
in the Chenar-e Olya area of Asadabad, Hamedan, Iran, using analytic signal, Euler deconvolution,
horizontal gradient and tilt derivative methods. Bollettino di Geofisica Teorica ed Applicata, 57,
329–342.
Mansouri, E., Feizi, F. and KarbalaeiRamezanali, A., 2015, Identification of magnetic
anomalies based on ground magnetic data analysis using multifractal modelling: a case study in
Qoja-Kandi, East Azerbaijan Province, Iran. Nonlinear Processes in Geophysics, 22, 579–587.
Mansouri, E., Feizi, F., 2016, Introducing Au potential areas, using remote  sensing and
geochemical data processing using fractal method in Chartagh, western Azarbijan – Iran, E.
Mansouri,  F.  Feizi, Arch. Min. Sci., Vol., No 2, 397–414.






















**Table 1.** Wavelength ranges and spatial resolutions of ASTER bands (Abrams, 2000).

| Module | VNIR | SWIR | TIR |
|---|---|---|---|
| **Spectral bandwidth (µm)** | Band 1 0.52 - 0.60 | Band 4 1.650 - 1.700 | Band 10 8.125 - 8.475 |
| | Band 2 0.63 - 0.69 | Band 5 2.145 - 2.185 | Band 11 8.475 - 8.825 |
| | Band 3 N 0.78 - 0.86 | Band 6 2.185 - 2.225 | Band 12 8.925 - 9.275 |
| | Band 3 B 0.78 - 0.86 (backward looking) | Band 7 2.235 - 2.285 | Band 13 10.25 - 10.95 |
| | | Band 8 2.295 - 2.395 | Band 14 10.95 - 11.65 |
| | | Band 9 2.360 - 2.430 | |
| **Spatial resolution (m)** | 15 | 30 | 90 |

**Table2.** Formula of regression models used for Aster satellite image bands

| Types of Regression | Number of coefficients | Formula |
|---|---|---|
| **First-Degree** | 15 | $Y_1 = a_0 + a_1 x_1 + a_2 x_2 + \cdots + a_{14} x_{14}$ |
| **First-Degree** | 106 | $Y_2 = Y_1 + a_{15}x_1x_2 + a_{16}x_1x_3 + \cdots + a_{27}x_1x_{14} + a_{28}x_2x_3 + a_{29}x_2x_4 + \cdots$ <br> $+ a_{39}x_2x_{14} + a_{40}x_3x_4 + a_{41}x_3x_5 + \cdots + a_{50}x_3x_{14}$ <br> $+ a_{51}x_4x_5 + \cdots + a_{60}x_4x_{14} + a_{61}x_5x_6 + \cdots + a_{69}x_5x_{14}$ <br> $+ a_{70}x_6x_7 + \cdots + a_{77}x_6x_{14} + a_{78}x_7x_8 + \cdots + a_{84}x_7x_{14}$ <br> $+ a_{85}x_8x_9 + \cdots + a_{90}x_8x_{14} + a_{91}x_9x_{10} + \cdots + a_{96}x_9x_{14}$ <br> $+ a_{97}x_{10}x_{11} + \cdots + a_{100}x_{10}x_{14}$ <br> $+ a_{101}x_{11}x_{12} + \cdots + a_{103}x_{11}x_{14} + a_{104}x_{12}x_{13} + a_{105}x_{12}x_{14}$ <br> $+ a_{106}x_{13}x_{14}$ |

**Table 3.** The values of the $R^2$ , $R^2_{adj}$ and p-value of ANOVA test of 2 multivariate regression

models

| Models | $R^2$ | $R^2_{adj}$ | p-value (ANOVA) |
|---|---|---|---|
| $Y_1$ | 0.738 | 0.715 | 0 |
| $Y_2$ | 0.847 | 0.829 | 0 |





290

**Table 4.** The calculated coefficients of regression models 1 and 2.

| | Model 1 | | Model 2 | |
| --- | --- | --- | --- | --- |
| **variables** | Coefficients ($a_i$) | variables | Coefficients ($a_i$) | |
| **CST** | 0.275 | CST | 0.677 | |
| $x_1$ | -0.01 | $x_1$ | -0.014 | |
| $x_2$ | -0.12 | $x_2$ | -0.019 | |
| $x_3$ | -0.019 | $x_3$ | -0.045 | |
| $x_4$ | 0.003 | $x_4$ | 0.022 | |
| $x_5$ | -0.006 | $x_5$ | -0.017 | |
| $x_6$ | -0.005 | $x_6$ | -0.001 | |
| $x_7$ | - | $x_7$ | - | |
| $x_8$ | -0.004 | $x_8$ | -0.02 | |
| $x_9$ | -0.005 | $x_9$ | -0.006 | |
| $x_{10}$ | 0.009 | $x_{10}$ | -0.014 | |
| $x_{11}$ | 0.005 | $x_{11}$ | 0.024 | |
| $x_{12}$ | 0.016 | $x_{12}$ | 0.024 | |
| $x_{13}$ | 0.002 | $x_{13}$ | 0.018 | |
| $x_{14}$ | 0.022 | $x_{14}$ | 0.036 | |
| - | - | $x_1 x_4$ | -0.0009 | |
| - | - | $x_1 x_6$ | -0.0002 | |
| - | - | $x_4 x_9$ | -0.0009 | |
| - | - | $x_7 x_8$ | 0.00082 | |

292

293





**Fig. 1. a)** The location of the Sarvian area in the Urumieh–Dokhtar magmatic belt, Iran **b)** Geological map of Sarvian area (scale 1:25000)



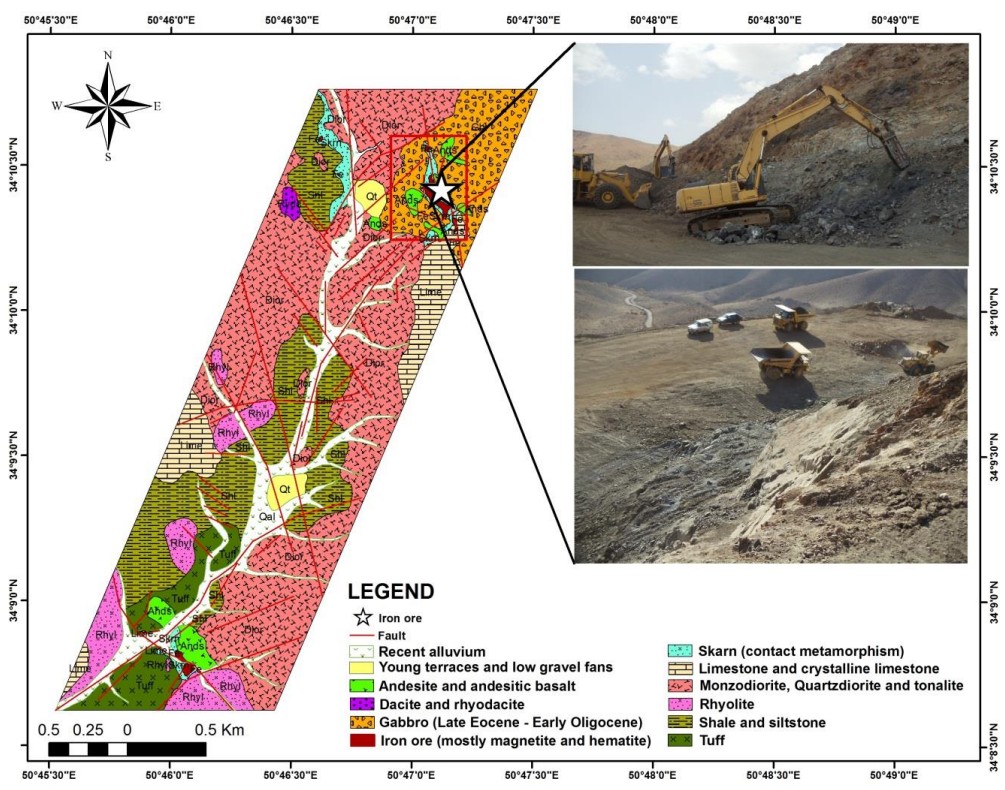


**Fig. 2.** Location of Sarvian iron mine in the study area




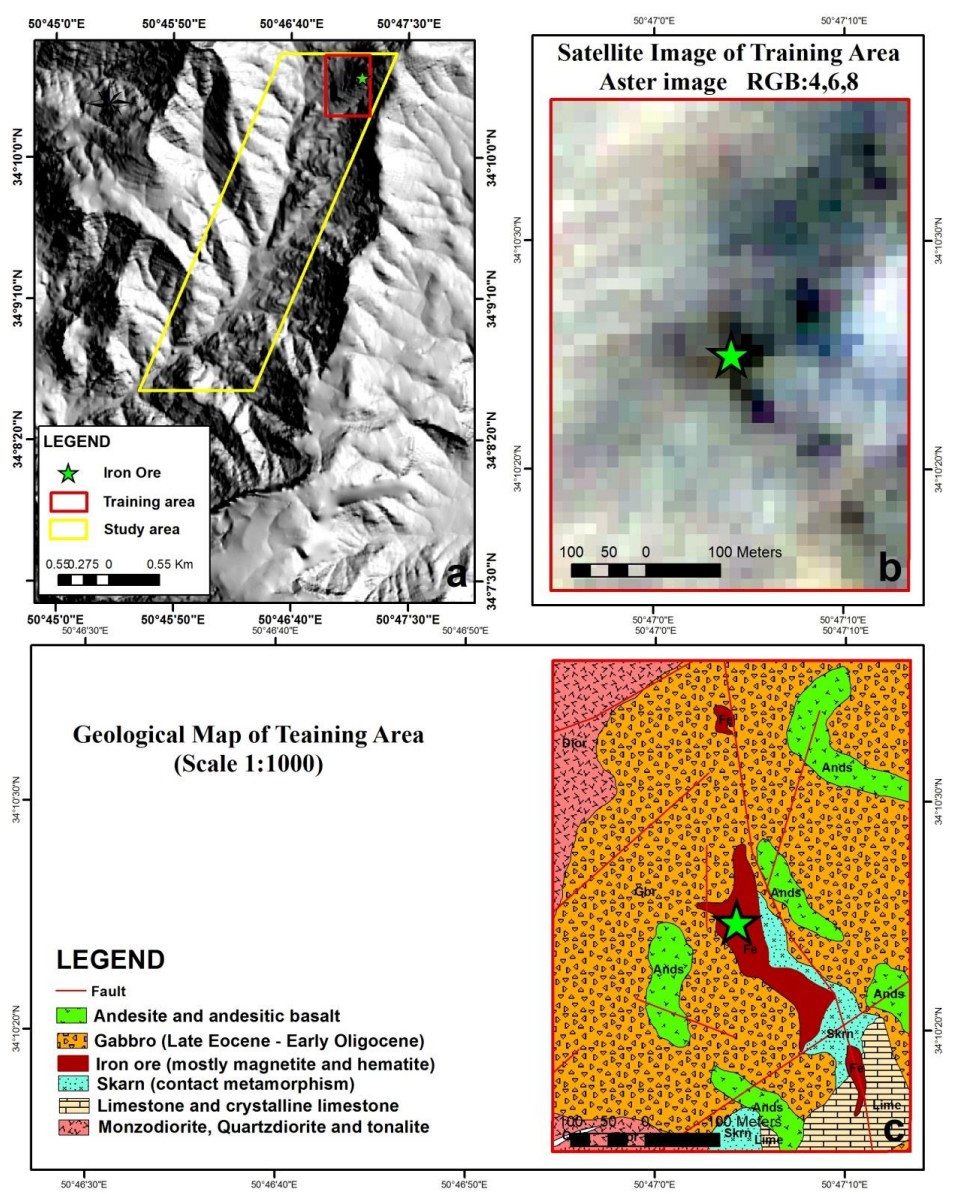


**Fig. 3.** a) Location of training area in the study area. b) Aster satellite image in the training area

302             (RGB:4,6,8). c) Geological map (scale 1:1000) of training area.





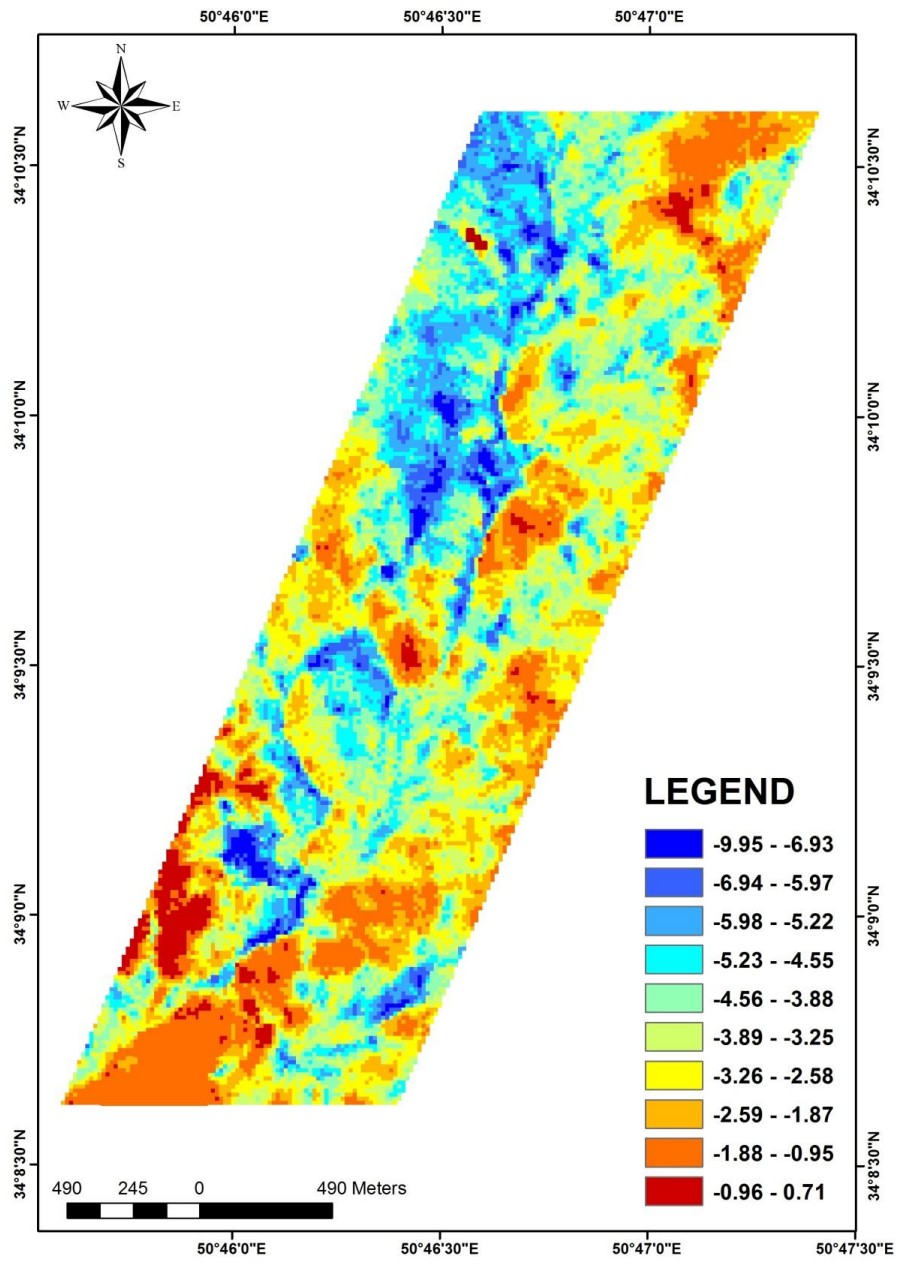

**Fig. 4.** Mineral prospectivity map of the Sarvian area.




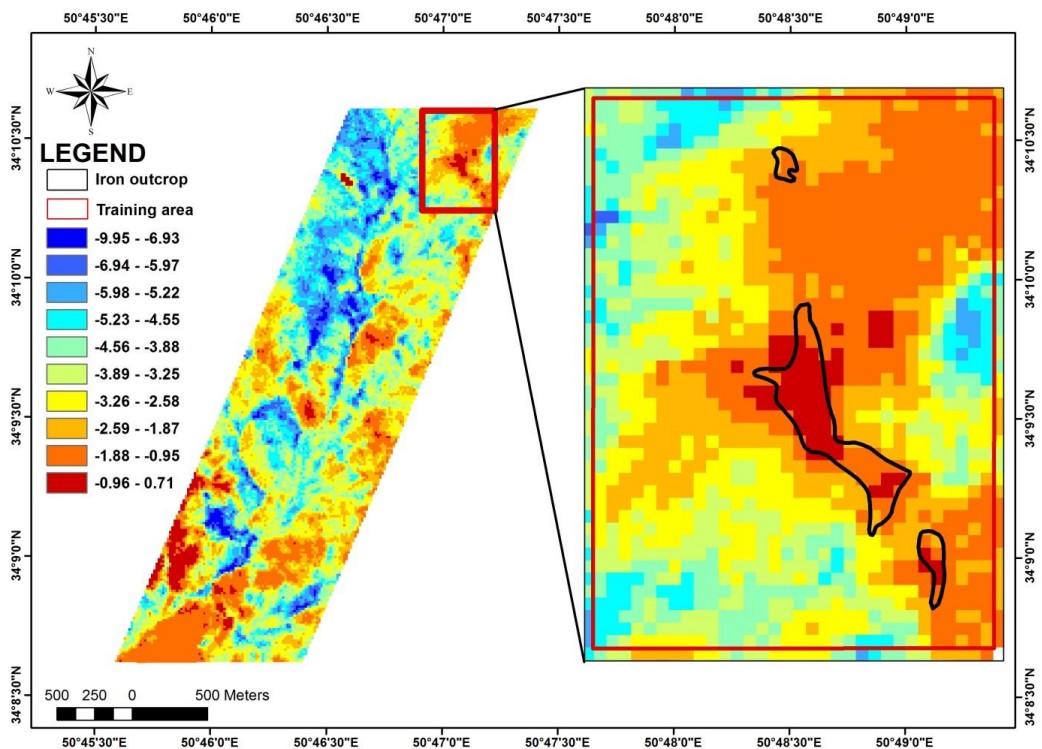


**Fig. 5.** Mineral prospectivity map of the Sarvian area which confirmed by iron outcrops.





**Fig. 6.** Mineral prospectivity map of the Sarvian area which confirmed by check field of three target

areas.