# Peer review of "Remote sensing data processing by multivariate regression analysis method for iron mineral resource potential mapping: A case study in Sarvian area, central Iran."

_Solid Earth, 2017_

## Short Comment (SC1) · 17 Mar 2017

I found this paper topical, relevant and interesting, as authors focus on the potential of Earth Observation (Remote Sensing) technique combined with Multivariate Regression Analyses. However, authors need to address a number of technical issues in this paper. I divided my observations to general observation and specific comments:

General observation: Introduction: I suggest a brief section that gives information on the relevance of remote sensing techniques in resource management with good references. To my understanding, the contribution of this paper to knowledge is on method-

ology. For me, the extensive discussion on Multivariate Regression Analyses gives a different impression to the paper though. This can be summarised.

Conclusion. The conclusion section basically focus on multivariate regression analysis (MRA). However, MRA should be a means to an end and not an end itself, please briefly discuss the results of your work in relation to the potential of MRA and remote sensing for iron mineral resource potential mapping as demonstrated for your study area.

References: I also observe that the referencing is not in conformity with the journal style. Please check this out.

Specific comment :

Line 79- 80. " Linear regression is used for modeling mineral prospetivity in the Sarvian area" . The statement is not appropriate here. It can go to literature review section..

Line 54-175: I suggest that you change this section to methodology [Study Area and Data collection and analysis] Line 155-173: What do you mean by regression analyses in the study area? I thought we should be looking at result and discussion section. You may wish to change this section to RESULT AND DISCUSSION.

Good luck !!!!

Please also note the supplement to this comment:
http://www.solid-earth-discuss.net/se-2017-25/se-2017-25-SC1-supplement.pdf

**Supplement:**

[revised manuscript text omitted]

---

## Referee Comment (RC1) · C. Pain (Referee) · 25 Mar 2017

Title: Remote sensing data processing by multivariate regression analysis method for iron mineral resource potential mapping: A case study in Sarvian area, central Iran Author(s): Edris Mansouri et al. Comments from Colin Pain

General comments This is an interesting paper that uses remotely sensed data supported by ground truth to prepare a map of iron mineral prospectivity.

Specific comments See the accompanying file. You have a lot of relevant references. However, I suggest a few more – these are from a simple search I carried out, and you

might like to include more. If you have trouble accessing them, please let me know. Xiaohui Li, Feng Yuan, Mingming Zhang, Cai Jia, Simon M. Jowitt, Alison Ord, Tongke Zheng, Xunyu Hu, Yang Li, Three-dimensional mineral prospectivity modeling for targeting of concealed mineralization within the Zhonggu iron orefield, Ningwu Basin, China, Ore Geology Reviews, Volume 71, December 2015, Pages 633-654. Renguang Zuo, Zhenjie Zhang, Daojun Zhang, Emmanuel John M. Carranza, Haicheng Wang, Evaluation of uncertainty in mineral prospectivity mapping due to missing evidence: A case study with skarn-type Fe deposits in Southwestern Fujian Province, China, Ore Geology Reviews, Volume 71, December 2015, Pages 502-515. Yihui Xiong, Renguang Zuo, Effects of misclassification costs on mapping mineral prospectivity, Ore Geology Reviews, Volume 82, April 2017, Pages 1-9.

Technical corrections The main problem with this paper is the very poor English expression. In the attached document I have made some suggested edits, and have highlighted, in yellow, sentences and phrases that I cannot understand. You must have the MS reviewed by an English language specialist. Also, the paper needs some slight reorganisation – see comments in the attached file, and also those from SC1.

Please also note the supplement to this comment:
http://www.solid-earth-discuss.net/se-2017-25/se-2017-25-RC1-supplement.pdf
* * *
[Figure]

**Supplement:**

(This file contains suggested edits, and some comments. Equations were left out for simplicity. Yellow highlights mark places where I cannot understand what is meant.)

Colin Pain

**Remote sensing data processing by multivariate regression analysis method for iron mineral resource potential mapping: A case study in the Sarvian area, central Iran**

Edris Mansouri[1], Faranak Feizi[2*], Alireza Jafari Rad[3], Mehran Arian[4]

1- Department of Geology, Science and Research Branch, Islamic Azad University, Tehran, Iran

2- Department of Mining Engineering, South Tehran Branch, Islamic Azad University, Tehran, Iran

3- Department of Geology, Science and Research branch, Islamic Azad University, Tehran, Iran

4- Department of Geology, Science and Research Branch, Islamic Azad University, Tehran, Iran

*corresponding author

Fax number: +98 021 88830012; Cell: +98 912 3006753; faranakfeizi@gmail.com.

**ABSTRACT**

This paper uses multivariate regression to create a mathematical model for iron skarn exploration in the Sarvian area, central Iran, using multivariate regression for Mineral Prospectivity Mapping (MPM). The main target of this manuscript is to apply multivariate regression analysis (as an MPM method) to mapping iron outcrops in the northeast part of the study area in order to discover new iron deposits in other parts of the study area. Two types of multivariate regression models using two linear equations were employed to discover new mineral deposits. Aster satellite images (14 bands) were used as Unique Independent Variables (UIVs), and iron outcrops were mapped as dependent variables for MPM. According to the results of p-value, $R^2$ and $R^2_{adj}$, the second regression model (which was a multiples and exponents of UIVs) fitted better than other models. The accuracy of the model was confirmed by iron outcrops map and geological observations. Based on field observation, iron mineralization occurs at the contact of limestone and intrusive rocks (skarn type). Iron minerals consist dominantly of magnetite, hematite and goethite.

**Key words:** Multivariate regression, Mineral Prospectivity Mapping, Iron, Sarvian

**1. INTRODUCTION**

Diagnosing futuristic zones and finding new mineral deposits in the region of interest, is the definitive main of mineral investigation. One way to achieve this aim is using satellite image processing for Mineral Prospectivity Mapping (MPM) (Carranza, 2008; Abedi et al., 2013; Golshadi et al., 2016 and Feizi et al., 2012).

The utilization of satellite images for mineral investigation has been extremely effective in indicating the presence of minerals. Likewise, remote sensing gives a synoptic view, which is useful for identifying and delineating different landscapes, linear features, and structural elements (Feizi and Mansouri, 2013b).

The main objective of this research was to use multivariate regression analysis (as a MPM method) to use pixel values from Aster satellite images of the north-east part of the study area to identify new iron deposits in other parts. Two types of multivariate regression models wereused to find new mineral deposits, using pixel values of Aster satellite image bands (14 bands) as Unique Independent Variables (UIVs), and iron outcrop areas (digitized a 1:5000geology map of the study area and field data as dependent variables.

Regression analyses have been utilized as a part of numerous logical fields, such as geosciences.

Multiple regression analyses have been used to identify stream sediment anomalies (e.g., Carranza, 2010a; Carranza, 2010b). Likewise, multivariate regression has been effectively used by Granian et al. (2015) to display subsurface mineralization from lithogeochemical information. Granian et al. (2015) used four types of multivariate regression models to depict significant surface geochemical anomalies indicating subsurface gold mineralization utilizing borehole data as dependent variables and surface lithogeochemical data as independent variables.

This paper uses multivariate regression to develop a useful and precise mathematical model of iron potential zones using remote sensing of the region of the interest.

**2. STUDY AREA**

The Sarvian area is in the Orumieh-Dokhtar magmatic arc in Central of Iran (Fig. 1a).

This magmatic arc is the most important metallogenic area inside the district and hosts large metal deposits such as lead, zinc, copper and iron (Feizi et al., 2016 and

Feizi et al., 2017).

The study area is dominated by Eocene intrusive rocks and carbonates of the Qom formation.

Several types of metal and non-metal mineral ore deposits have been reported in the study area. According to the 1:100,000 geological map of Kahak, the lithology of this area includes cream limestone with intercalations of marls (Qom formation), dark green, andesitic-basaltic lava, volcanic breccia, hyaloclastic limestone, green megaporphyritic andesitic-basaltic lava, rhyodacitic domes, tonalite-quartzdiorite, microquartzdiorite-microquartzmonzo-diorite, granite- granodiorite, altered light green, grey tuff, tuffaceous sandstone and shale with intercalation of nummulitic sandy limestone and andesitic lava, and orbitolina-bearing, thick bedded to massive grey limestone (Aptian-Albian) (Feizi et al., 2016) (Fig. 1b).

These relationships are demonstrated by the calcic iron skarn ore (Sarvian mine) in the northeast of study area (Feizi et al., 2017) (Fig. 2).

**3. METHODS AND DATA**

**3.1 Multivariate Regression**

Regression analyses is a good statistical tool for analysing relationships among dependent and independent variables (Granian et al., 2015).

In regression analyses, for dependent variables ($Y$) and independent variables ($x_i$), the equation is:

(1)

$Y$ can be a linear or non-linear function.

For linear regression Y is defined as follows:

(2)

For this function, the constant factor is $a_o$ , the random error is $\varepsilon$ , and the regression coefficients are $a_i$. If there are $n$ samples in a data set, for each sample $t$ variables were measured. Thus, function (2) is as follows:

                                    (3)

Equation (3) can be re-written as a matrix. The linear function matrix is:

                                    (4)

                                    (5)

The least squares technique is used for estimating [*A*] as the coefficient matrix, as follows:

                                    (6)

The inverse of variance-covariance samples matrix is $[\sum]^{-1}$ and the covariance matrix among independent variable and samples is [*C*]. Thus the regression coefficients model is calculated from equation 6.

The following criteria were used for the regression analysis:

1. The variance and the mean of the random error should be a constant value and zero, respectively.

2. The coefficient of determination value ($R^2$) should be examined. This value is calculated as follows (Granian et al., 2015):

                          (7)

The mean of the variable (Y), value of the $i$th sample ($Y_i$) and estimated value of the $i$th sample ($\hat{Y}$) for dependent variables were used in equation 7. The calculated $R^2$ value determined within

[0, 1] range. The value of $R^2$ is close to 1 for well fitted models.

1. Given the fact that adding independent variables to the model will increase the $R^2$

value, the adjusted determination coefficient ($R^2_{adj}$) is defined as follows (Granian et al.,

2015):

                          (8)

As it was mentioned, $n$ is number of samples (or data) and $t$ is the number of variables (or regression coefficients). If a set of explanatory variables are introduced into a regression one at a time, with the $R^2_{adj}$ computed each time, the level at which $R^2_{adj}$ reaches a maximum, and decreases afterward, would be a well fitted model.

2. In regression analyses, the model should be fitted to the data. Accordingly, the p-value of the regression model in the Analysis of variance (ANOVA) test should be acceptable (less than or equal to 0.05). Calculating the p-value of final coefficients for each model, may also help optimize and improve the model. This criterion could be applied after choosing the best model.

3.2. **Geo-data Preparation**

There are several iron ore bodies and one iron mine in the northeastern Sarvian study area.

The regional geological conditions of the area, suggest that the Sarvian iron mine is a good model for exploring the surrounding area. In this paper, a geology map of the mine is used as a training area for satellite imagery. In the training area, this method can model the iron outcrops (a dependent variable) based on Aster satellite image bands (independent variables) (Fig. 3).

**Figure 3 is about here.**

**3.2. REMOTE SENSING DATA (INDEPENDENT VARIABLES)**

The ASTER sensor was launched in December 1999 on board the Earth Observation System (EOS) US Terra satellite. ASTER

provides high-resolution images of the land surface, water, ice, and clouds using three separate sensor subsystems covering 14 multi-spectral bands from visible to thermal infrared (Table 1).

Resolutions are 15m, 30m, and 90m in the Visible and Near Infrared (VNIR), Shortwave Infrared (SWIR), and Thermal Infrared (TIR), respectively.

For more information see

Feizi and Mansouri, (2013b) and Mansouri and Feizi, (2016).

Several factors influence the signal measured at the sensor, for example, float of the sensor radiometric calibration, and atmospheric and topographical effects. In this way, Aster data were analysed using ENVI5.1 software to provide information such as wavelength, and log residuals which are basic for multispectral analyses (Mansouri et al., 2015).

In this study after correction, the pixel size of SWIR and TIR bands based on VNIR3 band (Panchromatic band) was converted to 15 m. The layer stacking function was then used to build a new multiband file from georeferenced images of various pixel sizes, extents, and projections.

**3.3. MAPPING OF IRON OUPCROPS (DEPENDENT VARIABLE)**

There are several iron veins and outcrops around the iron ore skarn mine in the north-eastern part of the Sarvian area. Iron outcrops in the training area were mapped using a geological map at a scale of 1:1000 of the iron ore deposit. The map was then field checked. The shape file layer of iron outcrops was converted to a raster file with a pixel size of 15 m.

**4. RESULTS OF REGRESSION ANALYSES**

Multiple, factorial, polynomial and response surface regressions have been utilized in many fields including the geosciences (e.g. Granian et al., 2015). In this study; Model 1 ($Y_1$) was generated as a multiple linear regression model and Model 2 ($Y_2$) was created from $Y_1$ plus many UIVs. The formulas for the two models are presented in Table 2. Thus, two linear equations ($Y_1$ and $Y_2$) were used to discover new mineral deposits, using pixel values from ASTER satellite data as independent variables and a map of iron outcrops as dependent variables. Of the two models proposed in this paper, model 2 has

106 coefficients (14 for UIVs, 1 as constant, 91 for multiples of UIVs) and model 1 has 15

coefficients (14 for UIVs, 1 as constant, 0 for multiples and exponents of UIVs) (Table 2).

Regression analyses were performed to assess the models in Table 2, and the critical criteria mentioned above, were examined. The values of the $R^2$, $R^2_{adj}$
* * *
Comment [CP2]: Is this what you mean?

and p-value from the ANOVA test of 2 multivariate regression models are provided in Table 3.

Table 4 presents the calculated coefficients of independent variables in regression models. Excluded independent variables are not mentioned in Table 4.

Excluded variables were those that had no effect on iron mineralization and the mapped distribution of iron outcrops.

5. **DISCUSSION**

We used several criteria to review the differences between the two models.

Firstly, the variance and the mean of the random error were acceptable for both models. Secondly, based on Table 4, the p-values of ANOVA test of the two models were equal to 0. For regression models the acceptable p-value should be less than or equal to 0.05. Thus, this criterion confirmed the validity of the models without specifying the most appropriate model.

The value of $R^2$ is close to 1 for well fitted models. The $R^2$ values of regression models are presented in Table 3. Model $Y_1$ has a lower $R^2$ than

$Y_2$. Thus, the $Y_2$ model is better than the $Y_1$ model.

Because adding independent variables to the model will increasing the $R^2$ value, the $R^2_{adj}$ value should be checked. The $R^2_{adj}$ values of regression models are presented in Table 3.

As mentioned above, if a set of variables are introduced into a regression, with the

$R^2_{adj}$ computed each time, the level at which $R^2_{adj}$ reaches a maximum, and decreases afterward, would be a well-fitted model. So, according to Table 3, $Y_2$ is the fitted model versus other models. Thus, $Y_2$ regression model is the most appropriate model for Mineral Prospectivity

Mapping.

Thus according to the results of p-value (ANOVA test), $R^2$ and $R^2_{adj}$, the second regression model ($Y_2$) would be the fitted model versus other models. For generating a mineral prospectivity map the model $Y_2$ was implemented in ArcGIS using the raster calculator tool. The normalized mineral prospectivity map of the study area is presented in Fig. 4.

To assess the accuracy of the selected model, the created prospectivity map was checked against the iron outcrops map in the northeast part of the study area (Fig. 5). The locations of iron outcrops is in close agreement with predictions from the mineral prospectivity map. In addition three target areas with very high potential were checked for iron outcrops and the prospectivity map was confirmed by geological observations (Fig. 6). Based on field observation iron mineralization occurs at contacts between limestone and intrusive rocks (skarn type). Iron mineralization consists dominantly of magnetite, hematite and goethite. Therefore, the accuracy of the mineral prospectivity map is confirmed in the Sarvian area.

**6. CONCLUSION**

The conclusions of this manuscript are as follows.

1) The application of multivariate regression analysis (as a MPM method) was confirmed in the Sarvian area. This paper used multivariate regression to create a mathematical model (with reasonable accuracy) for iron mineral exploration in the region of interest.

2) Two types of multivariate regression models, as two linear equations, were employed to discover new mineral deposits. According to the results of p-value, $R^2$ and $R^2_{adj}$,

, the second regression model best fitted obsservations.

3) The accuracy of the model was confirmed by iron outcrop mapping and geological observations. Based on field observation iron mineralization occurs in contacts between limestone and intrusive rocks (skarn type). Iron mineralization consists dominantly of magnetite, hematite and goethite.

**220 ACKNOWLEDGEMENTS**

The authors would like to thank Amirabbas Karbalaei Ramezanali for his helpful suggestions.

**224 REFRENCES**

Carranza, E.J.M., 2008, Geochemical anomaly and mineral prospectivity mapping in GIS,

Handbook of Exploration and Environmental Geochemistry. Elsevier, Amsterdam, 368 p.

Carranza, E.J.M., 2010a, Catchment basin modelling of stream sediment anomalies revisited:

incorporation of EDA and fractal analysis. Geochemistry: Exploration, Environment, Analysis,

10, 365-381.

Carranza, E.J.M., 2010b, Mapping of anomalies in continuous and discrete fields of stream sediment geochemical landscapes. Geochemistry: Exploration, Environment, Analysis, 10, 171-

187.

Feizi, F. and Mansouri, E., 2012, Identification of Alteration Zones with Using ASTER Data in a Part of Qom Province, Central Iran. Journal of Basic and Applied Scientific Research, 2, 73-

84.

Feizi, F. and Mansouri, E., 2013a, Separation of Alteration Zones on ASTER Data and

Integration with Drainage Geochemical Maps in Soltanieh, Northern Iran. Open Journal of

Geology, 3, 134-142.

Feizi, F. and Mansouri, E., 2013b, Introducing the Iron Potential Zones Using Remote Sensing

Studies in South of Qom Province, Iran. Open Journal of Geology, 3, 278-286.

Feizi, F., Mansouri, E. and KarbalaeiRamezanali, A., 2016, Prospecting of Au by Remote

Sensing and Geochemical Data Processing Using Fractal Modelling in Shishe-Botagh, Area (NW

Iran). Journal of the Indian Society of Remote Sensing, 44, 539-552.

Feizi, F., KarbalaeiRamezanali, A. and Mansouri, E., 2017, Calcic iron skarn prospectivity mapping based on fuzzy AHP method, a case Study in Varan area, Markazi province,

Iran. Geosciences Journal, 21, 123-136.

Granian, H., Tabatabaei, S. H., Asadi, H. H. and Carranza, E. J. M., 2015, Multivariate regression analysis of lithogeochemical data to model subsurface mineralization: a case study from the Sari Gunay epithermal gold deposit, NW Iran. Journal of Geochemical Exploration, 148, 249-

258.

Golshadi, Z., KarbalaeiRamezanali, A. and Kafaei, K., 2016, Interpretation of magnetic data in the Chenar-e Olya area of Asadabad, Hamedan, Iran, using analytic signal, Euler deconvolution, horizontal gradient and tilt derivative methods. Bollettino di Geofisica Teorica ed Applicata, 57,

329-342.

Mansouri, E., Feizi, F. and KarbalaeiRamezanali, A., 2015, Identification of magnetic anomalies based on ground magnetic data analysis using multifractal modelling: a case study in

Qoja-Kandi, East Azerbaijan Province, Iran. Nonlinear Processes in Geophysics, 22, 579-587.

Mansouri, E., Feizi, F., 2016, Introducing Au potential areas, using remote sensing and geochemical data processing using fractal method in Chartagh, western Azarbijan -Iran, E.

Mansouri, F. Feizi, Arch. Min. Sci., Vol., No 2, 397-414.

Figure captions

Fig. 1. a) The location of the Sarvian area in the Urumieh–Dokhtar magmatic belt, Iran b) Geological map of the Sarvian area (scale 1:25000).

Fig. 2. Location of the Sarvian iron mine in the study area

Fig. 3. a) Location of the training area, b) ASTER satellite image of the training area (RGB:4,6,8). c) Geological map (scale 1:1000) of the training area.

Fig. 4. Mineral prospectivity map of the Sarvian area.

Fig. 5. Mineral prospectivity map of the Sarvian area confirmed by iron outcrops.

Fig. 6. Mineral prospectivity map of the Sarvian area confirmed by check fields of the three target areas.

Comment [CP4]: Please explain th numbers in the legend.

---

## Author Comment (AC1) · 28 Mar 2017

Dear Editor I should say thank you very much on a behalf of the authors of "Remote sensing data processing by multivariate regression analysis method for iron mineral resource potential mapping: A case study in Sarvian area, central Iran." for your kindly praised and your nice recommendations. We added our new descriptions about your recommendations in red color in our manuscript and below. Kind regards, Faranak Feizi - I suggest a brief section that gives information on the relevance of remote sensing techniques in resource management with good references. Answer: some sen-

tences added for this comment.

- For me, the extensive discussion on Multivariate Regression Analyses gives a different impression to the paper though. This can be summarised. Answer: some sentences added for this comment.

- The conclusion section basically focus on multivariate regression analysis (MRA). However, MRA should be a means to an end and not an end itself, please briefly discuss the results of your work in relation to the potential of MRA and remote sensing for iron mineral resource potential mapping as demonstrated for your study area. Answer: some sentences added for this comment.

- I also observe that the referencing is not in conformity with the journal style. Please check this out. Answer: All references are corrected. - line 36-37-38. "The main objective of this manuscript is to use multivariate regression analysis (as a MPM method) to pixel values of Aster satellite image from north-east part of the study area to identify new iron deposits in other parts." I suggest that you mentioned this after the brief review of literature in this section. Answer: This part is corrected

- Line 79- 80. "Linear regression is used for modeling mineral prospetivity in the Sarvian area". The statement is not appropriate here. It can go to literature review section. Answer: This sentence is corrected. - Line 102. "In regression analysis, some criteria are required to review. These criteria are as follows:" This is not important here. Kindly remove. Answer: These parameters have calculated and presented in table 3 - Line 54-175: I suggest that you change this section to methodology [Study Area and Data collection and analysis]. Answer: All heading are corrected - Line 155-173: What do you mean by regression analyses in the study area? I thought we should be looking at result and discussion section. You may wish to change this section to RESULT AND DISCUSSION. Answer: This part is corrected.

Please also note the supplement to this comment:

http://www.solid-earth-discuss.net/se-2017-25/se-2017-25-AC1-supplement.pdf

**Supplement:**

**Remote sensing data processing by multivariate regression analysis method for iron mineral resource potential mapping: A case study in the Sarvian area, central Iran.**

Edris Mansouri[1], Faranak Feizi[2*], Alireza Jafari Rad[1], Mehran Arian[1]

1- Department of Geology, Science and Research branch, Islamic Azad University, Tehran, Iran

2- Department of Mining Engineering, South Tehran branch, Islamic Azad University, Tehran, Iran

*corresponding author: Faranak Feizi

Fax number: +98 021 88830012; Cell: +98 912 3006753; faranakfeizi@gmail.com.

**ABSTRACT**

This paper used multivariate regression to create a mathematical model (with reasonable accuracy) for iron skarn exploration in the region of the interest and generalizing multivariate regression in Mineral Prospectivity Mapping (MPM) field. The main target of this manuscript is to exert multivariate regression analysis (as a MPM method) to iron outcrops mapping from northeast part of the study area to discover new iron deposits in other parts. Two types of multivariate regression models as two linear equations were employed to discover new mineral deposits. The Aster satellite image bands (14 bands) sets as Unique Independent Variables (UIVs) and iron outcrops map as dependent variables were used for MPM. According to the results of p-value, $R^2$ and $R^2_{adj}$, the second regression model (which was a multiples and exponents of UIVs) was the fitted model versus other models. Also the accuracy of the model was confirmed by iron outcrops map and geological observations. Based on field observation iron mineralization occurs as contact of limestone and intrusive rocks (skarn type). Iron minerals consist dominantly of magnetite, hematite and goethite.

**Key words:** Multivariate regression, Mineral Prospectivity Mapping, Iron, Sarvian

**1. INTRODUCTION**

Preparing the information about an object without touching called remote sensing. The technology of acquiring data through a device which is located at a distance from the object and analysis of the data for interpreting the physical attributes of the object are two facts of remote sensing (Gupta, 2003).Recently, use of remotely-sensed data in natural resources mapping has been popular. In the other words, applications of remote sensing in geological investigations is

the best approach for large scale study (Melesse et al., 2007). In this research, we present some of the most commonly used applications of the techniques in mineral resources mapping.

Diagnosing futuristic zones and finding new mineral deposits in the region of interest, is the definitive main of mineral investigation. One way to achieve this aim is Using satellite image processing for identify Mineral Prospectivity Mapping (MPM) (Carranza, 2008; Abedi et al., 2013; Golshadi et al., 2016 and Feizi et al., 2012).

The utilization of satellite images for mineral investigation has been extremely effective in indicating out the attendance of minerals. Likewise, remote sensing gives the synoptic view, which is useful in distinguishing proof and delineation of different land frames, linear features, and structural elements (Feizi and Mansouri, 2013a).

The regression analysis is a statistical process for estimating the relationships among variables. There are many techniques for analyzing several variables, when the focus is on the relationship between a dependent variable and one or more independent variables, that the latter case, called multivariate regression analysis. This regression analyses have been utilized as a part of numerous logical fields, such as follow geoscience branches.

Identification of stream sediment anomalies have been used by multiple regression analyses (e.g., Carranza, 2010a; Carranza, 2010b). Likewise, multivariate regression has been effectively utilized by Granian et al. (2015) to display subsurface mineralization from lithogeochemical information. Granian et al. (2015) utilized four types of multivariate regression models to depict significant surface geochemical anomalies for acknowledgment subsurface gold mineralization utilizing borehole data as dependent variables and surface lithogeochemical data as independent variables.

Based on previous work such as Allbed et al., (2012), modelling and mapping satellite images data based on regression analysis and remote sensing data is a promising approach, as it facilitates timely detection with a low-cost procedure and allows decision makers to decide what necessary action should be taken in the early stages to Mineral Prospectivity Mapping (MPM) field.

[revised manuscript text omitted]
 regression analysis is an appropriate and direct method for MPM by satellite images data. In this paper, the output of processed satellite image using regression analysis indicates the iron potential zones accurately.

2) The application of multivariate regression analysis (as a MPM method) was confirmed in the Sarvian area. This paper used multivariate regression to create a mathematical model (with reasonable accuracy) for iron mineral exploration in the region of the interest and generalizing multivariate regression in MPM field.

3) Two types of multivariate regression models as two linear equations were employed to discover new mineral deposits. According to the results of p-value, $R^2$ and $R^2_{adj}$, the second regression model was the fitted model versus other models.

4) Also the accuracy of the model was confirmed by iron outcrops map and geological observations. Based on field observation iron mineralization occurs contact of limestone and intrusive rocks (skarn type). Iron mineralizations consist dominantly of magnetite, hematite and goethite.

5) The results demonstrate that modelling and mapping satellite images data based on regression analysis and remote sensing data is an efficient approach, as it facilitates timely detection with a low-cost procedure and allows decision makers to decide what necessary action should be taken in the early stages to Mineral Prospectivity Mapping (MPM) field.

6) Regression analysis method is a subset of supervised classification due to the mentioned procedure. In this method, target spectrums of training area are used for modeling and MPM.

**ACKNOWLEDGEMENTS**

The authors would like to thank Amirabbas KarbalaeiRamezanali for his helpful suggestions.

**REFRENCES**

Abedi, M., Torabi, S.A. and Norouzi, G.H.: Application of fuzzy AHP method to integrate geophysical data in a prospect scale, a case study: Seridune copper deposit. Bollettino di Geofisica Teorica, 54, 145–164, 2013.

Abrams, M.: The Advanced Spaceborne Thermal Emission and Reflection Radiometer (ASTER): Data products for the high spatial resolution imager on NASA's Terra platform, International Journal of Remote Sensing, 21, 5, 847-859, 2000.

Carranza, E.J.M.: Geochemical anomaly and mineral prospectivity mapping in GIS, Handbook of Exploration Environmental Geochemistry. Elsevier, Amsterdam, 368 p, 2008.

Carranza, E.J.M.: Catchment basin modelling of stream sediment anomalies revisited: incorporation of EDA and fractal analysis. Geochemistry: Exploration, Environment, Analysis, 10, 365–381, 2010a.

Carranza, E.J.M.: Mapping of anomalies in continuous and discrete fields of stream sediment geochemical landscapes. Geochemistry: Exploration, Environment, Analysis, 10, 171–187, 2010b.

Feizi, F. and Mansouri, E.: Identification of Alteration Zones with Using ASTER Data in A Part of Qom Province, Central Iran. Journal of Basic and Applied Scientific Research, 2, 73–84, 2012.

Feizi, F. and Mansouri, E.: Separation of Alteration Zones on ASTER Data and Integration with Drainage Geochemical Maps in Soltanieh, Northern Iran. Open Journal of Geology, 3, 134–142, 2013a.

Feizi, F. and Mansouri, E.: Introducing the Iron Potential Zones Using Remote Sensing Studies in South of Qom Province, Iran. Open Journal of Geology, 3, 278–286, 2013b.

Feizi, F., Mansouri, E. and KarbalaeiRamezanali, A.: Prospecting of Au by Remote Sensing and Geochemical Data Processing Using Fractal Modelling in Shishe-Botagh, Area (NW Iran). Journal of the Indian Society of Remote Sensing, 44, 539–552, 2016.

Feizi, F., KarbalaeiRamezanali, A. and Mansouri, E.: Calcic iron skarn prospectivity mapping based on fuzzy AHP method, a case Study in Varan area, Markazi province, Iran. Geosciences Journal, 21, 123–136, 2017.

Granian, H., Tabatabaei, S. H., Asadi, H. H. and Carranza, E. J. M.: Multivariate regression analysis of lithogeochemical data to model subsurface mineralization: a case study from the Sari Gunay epithermal gold deposit, NW Iran. Journal of Geochemical Exploration, 148, 249–258, 2015.

Golshadi, Z., KarbalaeiRamezanali, A. and Kafaei, K.: Interpretation of magnetic data in the Chenar-e Olya area of Asadabad, Hamedan, Iran, using analytic signal, Euler deconvolution, horizontal gradient and tilt derivative methods. Bollettino di Geofisica Teorica ed Applicata, 57, 329–342, 2016.

Gupta, R. P.: Remote sensing geology, Berlin, Heidelberg: Springer Berlin Heidelberg: Imprint: Springer, 2003.

Allbed, A., Kumar, L., and Sinha, P.: Mapping and Modelling Spatial Variation in Soil Salinity in the Al Hassa Oasis Based on Remote Sensing Indicators and Regression Techniques, Remote Sens. 6, 1137-1157; 2014.

Mansouri, E., Feizi, F. and KarbalaeiRamezanali, A.: Identification of magnetic anomalies based on ground magnetic data analysis using multifractal modelling: a case study in Qoja-Kandi, East Azerbaijan Province, Iran. Nonlinear Processes in Geophysics, 22, 579–587, 2015.

Mansouri, E., Feizi, F.: Introducing Au potential areas, using remote  sensing and geochemical data processing using fractal method in Chartagh, western Azarbijan – Iran, E. Archive of  Mining Sciences, Vol., No 2, 397–414, 2016.

Melesse, A. M., Weng, Q., Thenkabail, P. S. and Senay, G. B.: Remote Sensing Sensors and Applications in Environmental Resources Mapping and Modelling, Sensors, 7, 3209-3241, 2007.

**Table 1.** Wavelength ranges and spatial resolutions of ASTER bands (Abrams, 2000).

| Module | VNIR | SWIR | TIR |
|---|---|---|---|
| | Band 1 0.52 - 0.60 | Band 4 1.650 - 1.700 | Band 10 8.125 - 8.475 |
| | Band 2 0.63 - 0.69 | Band 5 2.145 - 2.185 | Band 11 8.475 - 8.825 |
| | Band 3 N 0.78 - 0.86 | Band 6 2.185 - 2.225 | Band 12 8.925 - 9.275 |
| **Spectral bandwidth (µm)** | Band 3 B 0.78 - 0.86 (backward looking) | Band 7 2.235 - 2.285 | Band 13 10.25 - 10.95 |
| | | Band 8 2.295 - 2.395 | Band 14 10.95 - 11.65 |
| | | Band 9 2.360 - 2.430 | |
| **Spatial resolution (m)** | 15 | 30 | 90 |

**Table2.** Formula of regression models used for Aster satellite image bands

| Types of Regression | Number of coefficients | Formula |
|---|---|---|
| First-Degree | 15 | $Y_1 = a_0 + a_1 x_1 + a_2 x_2 + \cdots + a_{14} x_{14}$ |
| First-Degree | 106 | $Y_2 = Y_1 + a_{15} x_1 x_2 + a_{16} x_1 x_3 + \cdots + a_{27} x_1 x_{14} + a_{28} x_2 x_3 + a_{29} x_2 x_4 + \cdots$ $+ a_{39} x_2 x_{14} + a_{40} x_3 x_4 + a_{41} x_3 x_5 + \cdots + a_{50} x_3 x_{14}$ $+ a_{51} x_4 x_5 + \cdots + a_{60} x_4 x_{14} + a_{61} x_5 x_6 + \cdots + a_{69} x_5 x_{14}$ $+ a_{70} x_6 x_7 + \cdots + a_{77} x_6 x_{14} + a_{78} x_7 x_8 + \cdots + a_{84} x_7 x_{14}$ $+ a_{85} x_8 x_9 + \cdots + a_{90} x_8 x_{14} + a_{91} x_9 x_{10} + \cdots + a_{96} x_9 x_{14}$ $+ a_{97} x_{10} x_{11} + \cdots + a_{100} x_{10} x_{14}$ $+ a_{101} x_{11} x_{12} + \cdots + a_{103} x_{11} x_{14} + a_{104} x_{12} x_{13} + a_{105} x_{12} x_{14}$ $+ a_{106} x_{13} x_{14}$ |

**Table 3.** The values for $R^2$, $R^2_{adj}$ and p-value of ANOVA test of 2 multivariate regression models

| Models | $R^2$ | $R^2_{adj}$ | p-value (ANOVA) |
|---|---|---|---|
| $Y_1$ | 0.738 | 0.715 | 0 |
| $Y_2$ | 0.847 | 0.829 | 0 |

**Table 4.** The calculated coefficients of regression models 1 and 2.

| | Model 1 | | Model 2 | |
| --- | --- | --- | --- | --- |
| **variables** | Coefficients ($a_i$) | variables | Coefficients ($a_i$) | |
| **CST** | 0.275 | CST | 0.677 | |
| $x_1$ | -0.01 | $x_1$ | -0.014 | |
| $x_2$ | -0.12 | $x_2$ | -0.019 | |
| $x_3$ | -0.019 | $x_3$ | -0.045 | |
| $x_4$ | 0.003 | $x_4$ | 0.022 | |
| $x_5$ | -0.006 | $x_5$ | -0.017 | |
| $x_6$ | -0.005 | $x_6$ | -0.001 | |
| $x_7$ | - | $x_7$ | - | |
| $x_8$ | -0.004 | $x_8$ | -0.02 | |
| $x_9$ | -0.005 | $x_9$ | -0.006 | |
| $x_{10}$ | 0.009 | $x_{10}$ | -0.014 | |
| $x_{11}$ | 0.005 | $x_{11}$ | 0.024 | |
| $x_{12}$ | 0.016 | $x_{12}$ | 0.024 | |
| $x_{13}$ | 0.002 | $x_{13}$ | 0.018 | |
| $x_{14}$ | 0.022 | $x_{14}$ | 0.036 | |
| - | - | $x_1 x_4$ | -0.0009 | |
| - | - | $x_1 x_6$ | -0.0002 | |
| - | - | $x_4 x_9$ | -0.0009 | |
| - | - | $x_7 x_8$ | 0.00082 | |

[Figure]

**Fig. 1. a)** The location of the Sarvian area in the Orumieh–Dokhtar magmatic belt, Iran **b)** Geological map of the Sarvian area (scale 1:25000).

[Figure]

**Fig. 2.** Location of the Sarvian iron mine in the study area.

[Figure]

**Fig. 3.** a) Location of training area in the study area. b) Aster satellite image in the training area

(RGB:4,6,8). c) Geological map (scale 1:1000) of training area.

[Figure]

**Fig. 4.** Mineral prospectivity map of the Sarvian area.

[Figure]

**Fig. 5.** Mineral prospectivity map of the Sarvian area which confirmed by iron outcrops.

[Figure]

**Fig. 6.** Mineral prospectivity map of the Sarvian area which confirmed by check field of three target areas.

---

## Author Comment (AC2) · 4 Apr 2017

**Remote sensing data processing by multivariate regression analysis method for iron mineral resource potential mapping: A case study in the Sarvian area, central Iran**

Edris Mansouri[1], Faranak Feizi[2*], Alireza Jafari Rad[1], Mehran Arian[1]

1- Department of Geology, Science and Research branch, Islamic Azad University, Tehran, Iran

2- Department of Mining Engineering, South Tehran branch, Islamic Azad University, Tehran, Iran

*corresponding author: Faranak Feizi

Fax number: +98 021 88830012; Cell: +98 912 3006753; faranakfeizi@gmail.com.

**ABSTRACT**

This paper uses multivariate regression to create a mathematical model for iron skarn exploration in the Sarvian area, central Iran, using multivariate regression for Mineral Prospectivity Mapping (MPM). The main target of this manuscript is to apply multivariate regression analysis (as a MPM method) to mapping iron outcrops in the northeastern part of the study area in order to discover new iron deposits in other parts of the study area. Two types of multivariate regression models using two linear equations were employed to discover new mineral deposits. Aster satellite images (14 bands) were used as Unique Independent Variables (UIVs), and iron outcrops were mapped as dependent variables for MPM. According to the results of p-value, $R^2$ and $R^2_{adj}$, the second regression model (which was a multiples of UIVs) fitted better than other models. The accuracy of the model was confirmed by iron outcrops map and geological observations. Based on field observation, iron mineralization occurs at the contact of limestone and intrusive rocks (skarn type). Iron minerals consist dominantly of magnetite, hematite and goethite.

**Key words:** Multivariate regression, Mineral Prospectivity Mapping, Iron, Sarvian

**1. INTRODUCTION**

Preparing the information on an object without touching is called remote sensing. The technology of acquiring data through a device which is located at a distance from the object and the analysis of the data for the purpose of interpreting the physical attributes of the object are two facts of remote sensing (Gupta, 2003).Recently, application of remotely-sensed data in natural resources mapping has been popular. In the other words, applications of remote sensing in geological investigations is the best approach for large scale studies (Melesse et al., 2007). In this

research, we present some of the most commonly used applications of the techniques in mineral resources mapping.

Mineral exploration is a complicated process that involves a focus on delineation of target areas in the search for new mineral deposits (Xiong et al., 2017). The principal aim of mineral investigation in the region of interest is to diagnose futuristic zones and to find new mineral deposits. One way to achieve this aim is using satellite image processing in order to identify Mineral Prospectivity Mapping (MPM) (Carranza, 2008; Abedi et al., 2013; Golshadi et al., 2016 and Feizi et al., 2012).

Mineral prospectivity mapping using 2D geoscientific data, such as geological, geochemical, geophysical and remote sensing data, among others, has been widely used for mineral exploration and targeting for the last 30 years (Li et al., 2015, Abedi et al., 2012; Bonham-Carter and Agterberg, 1990; Carranza, 2009; Carranza and Sadeghi, 2010c; Ford and Blenkinsop, 2008; Lindsay et al., 2014; Lisitsin et al., 2013; Pan and Harris, 2000; Porwal et al., 2010).

The utilization of satellite images for mineral investigation has been extremely effective in indicating out the presence of minerals. Likewise, remote sensing provides a synoptic view, which is useful for identifying and delineation different landscapes, linear features, and structural elements (Feizi and Mansouri, 2013a).

The regression analysis is a statistical process in order to estimating the relationships among variables. There are many techniques for analyzing several variables, when the focus is on the relationship between a dependent variable and one or more independent variables, which the latter case is called multivariate regression analysis. This regression analyses has been utilized as part of numerous logical fields, such as geoscience branches.

Identification of stream sediment anomalies have been used by multiple regression analyses (e.g., Carranza, 2010a; Carranza, 2010b). Likewise, multivariate regression has been effectively utilized by Granian et al. (2015) to display subsurface mineralization from lithogeochemical information. Granian et al. (2015) used four types of multivariate regression models to depict significant surface geochemical anomalies indicating subsurface gold mineralization utilizing borehole data as dependent variables and surface lithogeochemical data as independent variables.

Based on previous work such as Allbed et al., (2012), modelling and mapping of mineral potentials based on satellite image data and processing it based on remote sensing and regression analysis is a promising approach as it facilitates timely detection with a low-cost procedure and allows decision makers to decide what necessary action should be taken as the first step in Mineral Prospectivity Mapping (MPM) field.

The main objective of this research was to use multivariate regression analysis (as a MPM method) to use pixel values from Aster satellite images of the northeastern part of the study area to identify new iron deposits in other parts. Two types of multivariate regression models were used to find new mineral deposits, using pixel values of Aster satellite image bands (14 bands) as Unique Independent Variables (UIVs), and iron outcrop areas (digitized a 1:5000 geology map of the study area and field) data as dependent variables.

This paper uses multivariate regression to develop a useful and precise mathematical model of iron potential zones the region of the interest.

**2. METHODOLOGY**

**2.1. STUDY AREA**

The Sarvian area is in the Orumieh-Dokhtar magmatic arc in Central Iran (Fig. 1a). This magmatic arc is the most important metallogenic area inside the district and hosts large metal deposits such as lead, zinc, copper and iron (Feizi et al., 2016 and Feizi et al., 2017).

The study area is dominated by Eocene intrusive rocks and carbonates of the Qom formation. Several types of metal and non-metal mineral ore deposits have, as of now, been reported in the study area. According to the 1:100,000 geological map of Kahak, the lithology of this area includes cream limestone with intercalations of marls (Qom formation), dark green, andesitic-basaltic lava, volcanic breccia, hyaloclastic limestone, green megaporphyritic andesitic-basaltic lava, rhyodacitic domes, tonalite-quartzdiorite, microquartzdiorite-microquartzmonzo-diorite, granite-granodiorite, altered of light green, grey tuff, tuffaceous sandstone and shale with intercalation of nummulitic sandy limestone and andesitic lava, and orbitolina-bearing, thick bedded to massive grey limestone (Aptian–Albian) (Feizi et al., 2016) (Fig. 1b).

**Figure 1 is about here.**

These relationships are demonstrated by the calcic iron skarn ore (Sarvian mine) in the northeast of study area (Feizi et al., 2017) (Fig. 2). Skarn-type Fe mineralization and alteration are localized along the contact zone between intrusive rocks and carbonate sequences (Zuo et al., 2014).

**Figure 2 is about here.**

**2.2. MULTIVARIATE REGRESSION**

Regression analyses is a good statistical tool for analyzing relationships among dependent and independent variables (Granian et al., 2015). In regression analyses, for dependent variables ($Y$) and independent variables called ($x_i$), the equation is:

$$Y = f(x_i). \qquad (1)$$

Y can be a linear or non-linear function. For linear regression Y is defined as follows:

$$Y = a_0 + a_1x_1 + a_2x_2 + \cdots + a_ix_i + \varepsilon, \quad i = 1,2,\dots,n. \quad (2)$$

For this function, the constant factor is $a_0$, the random error is $\varepsilon$, and the regression coefficients are $a_i$. If there are $n$ samples in a data set, for each sample $t$ variables were measured. Thus, function (2) is as follows:

$$Y_i = \hat{a}_0 + \hat{a}_1X_{i1} + \hat{a}_2X_{i2} + \cdots + \hat{a}_tX_{it} + \varepsilon_i i = 1,2,\dots,n. \quad (3)$$

Equation (3) can be re-written as a matrix. The linear function matrix is:

$$[Y] = [X][A] + [\varepsilon]. \quad (4)$$

$$[Y] = \begin{bmatrix} Y_1 \\ Y_2 \\ \cdot \\ \vdots \\ Y_n \end{bmatrix}; \ [A] = \begin{bmatrix} \hat{a}_0 \\ \hat{a}_1 \\ \cdot \\ \vdots \\ \hat{a}_t \end{bmatrix}; \ [X] = \begin{bmatrix} 1 & X_{11}X_{12}\dots X_{1t} \\ 1 & X_{21}X_{22}\dots X_{2t} \\ & \cdot \\ & \vdots \\ 1 & X_{n1}X_{n2}\dots X_{nt} \end{bmatrix}; \ [\varepsilon] = \begin{bmatrix} \varepsilon_1 \\ \epsilon_2 \\ \cdot \\ \vdots \\ \epsilon_n \end{bmatrix}. \quad (5)$$

The least squares technique is used for estimating $[A]$ as the coefficient matrix, as follows:

$$[A] = [\Sigma\ ]^{-1}[C] = ([X]'[X])^{-1}[X]'[Y]. \quad (6)$$

The inverse of variance-covariance samples matrix is $[\Sigma\ ]^{-1}$ and the covariance matrix among independent variable and samples is $[C]$. Thus the regression coefficients model is calculated from equation 6.

The following criteria were used for the regression analysis:

1. The variance and the mean of the random error should be a constant value and zero, respectively.

2. The coefficient of determination value ($R^2$) should be examined. This value is calculated as follows (Granian et al., 2015):

$$R^2 = \frac{\sum_{i=1}^{n}(\widehat{Y}_i - \bar{Y})^2}{\sum_{i=1}^{n}(Y_i - \bar{Y})^2} = 1 - \frac{\sum_{i=1}^{n}(Y_i - \widehat{Y}_i)^2}{\sum_{i=1}^{n}(Y_i - \bar{Y})^2}. \qquad (7)$$

The mean of the variable ($\bar{Y}$), value of the $i$th sample ($Y_i$) and estimated value of the $i$th sample ($\widehat{Y}_i$) for dependent variables were used in equation 7. The calculated $R^2$ value determined within [0, 1] range. The value of $R^2$ is close to 1 for well fitted models.

1. Given the fact that adding independent variables to the model will increase the $R^2$ value, the adjusted determination coefficient ($R^2_{adj}$) is defined as follows (Granian et al., 2015):

$$R^2_{adjusted} = 1 - \frac{n-1}{n-t}(1 - R^2). \qquad (8)$$

As it was mentioned, $n$ is number of samples (or data) and $t$ is the number of variables (or regression coefficients). If a set of explanatory variables are introduced into a regression one at a time, with the $R^2_{adj}$ computed each time, the level at which $R^2_{adj}$ reaches a maximum, and decreases afterward, would be a well fitted model.

2. In regression analyses, the model should be fitted to the data. Accordingly, the p-value of the regression model in the analysis of variance (ANOVA) test should be acceptable (less than or equal to 0.05). Calculating the p-value of final coefficients for each model, may also help optimize and improve the model. This criterion could be applied after choosing the best model.

**3. DATA COLLECTION**

There are several iron ore bodies and one iron mine in the northeastern Sarvian study area. The regional geological conditions of the area, suggest that the Sarvian iron mine is a good model for exploring the surrounding area. In this paper, a geology map of the mine is used as a training area for satellite imagery. In the training area, this method can model the iron outcrops (a dependent variable) based on Aster satellite image bands (independent variables) (Fig. 3).

**Figure 3 is about here.**

**3.1. REMOTE SENSING DATA (INDEPENDENT VARIABLES)**

The ASTER sensor was launched in December 1999 on board the Earth Observation System (EOS) US Terra satellite. ASTER provides high-resolution images of the land surface, water, ice, and clouds using three separate sensor subsystems covering 14 multi-spectral bands from visible to thermal infrared (Table 1). Resolutions are 15m, 30m, and 90m in the Visible and Near Infrared (VNIR), Shortwave Infrared (SWIR), and Thermal Infrared (TIR), respectively. For more information see Feizi and Mansouri, (2013b) and Mansouri and Feizi, (2016).

Several factors influence the signal measured at the sensor, for example, radiometric calibration, and atmospheric and topographical effects. In this way, Aster data were analysed using ENVI5.1 software to provide information such as wavelength, dark subtract and log residuals which are basic for multispectral analyses (Mansouri et al., 2015).

In this study after corrections, the pixel size of SWIR and TIR bands based on VNIR3 band (Panchromatic band) was converted to 15 m. The layer stacking function was then used to build a new multiband file from georeferenced images of various pixel sizes, extents, and projections.

**Table 1 is about here.**

**3.2. MAPPING OF IRON OUPCROPS (DEPENDENT VARIABLE)**

There are several iron veins and outcrops around the iron ore skarn mine in the north-eastern part of the Sarvian area. Iron outcrops in the training area were mapped using a geological map at a scale of 1:1000 of the iron ore deposit. The map was then field checked. The shape file layer of iron outcrops was converted to a raster file with a pixel size of 15 m.

**4. RESULTS AND DISCUSSION**

Multiple, factorial, polynomial and response surface regressions have been utilized in many fields including the geosciences (e.g. Granian et al., 2015). In this study; Model 1 ($Y_1$) was generated as a multiple linear regression model and Model 2 ($Y_2$) was created from $Y_1$ plus many UIVs. The formulas for the two models are presented in Table 2. Thus, two linear equations ($Y_1$ and $Y_2$) were used to discover new mineral deposits, using pixel values from ASTER satellite data as independent variables and a map of iron outcrops as dependent variables. Of The two models proposed in this paper, model 2 has 106 coefficients (14 for UIVs, 1 as constant, 91 for multiples of UIVs) and model 1 has 15 coefficients (14 for UIVs, 1 as constant, 0 for multiples and exponents of UIVs) (Table 2).
**Table 2 is about here.**

Regression analyses were performed to assess the models in Table 2, and the critical criteria mentioned above, were examined. The values of the $R^2$ , $R^2_{adj}$ and p-value from the ANOVA test of 2 multivariate regression models are provided in Table 3.
**Table 3 is about here.**

Table 4 presents the calculated coefficients of independent variables in regression models. Excluded independent variables are not mentioned in Table 4. Excluded variables were those that had no effect on iron mineralization and the mapped distribution of iron outcrops.

**Table 4 is about here.**

We used several criteria to review the differences between the two models. Firstly, the variance and the mean of the random error were acceptable for both models. Secondly, based on Table 4, the p-values of ANOVA test of the two models were equal to 0. For regression models, the acceptable p-value should be less than or equal to 0.05. Thus, this criterion confirmed the validity of the models without specifying the most appropriate model.

The value of $R^2$ is close to 1 for well fitted models. The $R^2$ values of regression models are presented in Table 3. Model Y1 has a lower $R_2$ than $Y_2$. Thus, the $Y_2$ model is better than the $Y_1$ model.

Because adding independent variables to the model will increasing the $R^2$ value, the $R^2_{adj}$ value should be checked. The $R^2_{adj}$ values of regression models are presented in Table 3. As mentioned above, if a set of variables are introduced into a regression, with the $R^2_{adj}$ computed each time, the level at which $R^2_{adj}$ reaches a maximum, and decreases afterward, would be a well-fitted model. So, according to Table 3, $Y_2$ is the fitted model versus other models. Thus, $Y_2$ regression model is the most appropriate model for Mineral Prospectivity Mapping.

Thus according to the results of p-value (ANOVA test), $R^2$ and $R^2_{adj}$, the second regression model ($Y_2$) would be the fitted model versus other models. For generating a mineral prospectivity map the model $Y_2$ was implemented in ArcGIS using the raster calculator tool. The normalized mineral prospectivity map of the study area is presented in Fig. 4.

**Figure 4 is about here.**

To assess the accuracy of the selected model, the created prospectivity map was checked against the iron outcrops map in the northeastern part of the study area (Fig. 5). The locations of iron outcrops is in close agreement with predictions from the mineral prospectivity map. In addition three target areas with very high potential were checked for iron outcrops and the prospectivity map was confirmed by geological observations (Fig. 6). Based on field observation iron mineralization occurs at contacts between limestone and intrusive rocks (skarn type). Iron mineralizations consists dominantly of magnetite, hematite and goethite. Therefore, the accuracy of the mineral prospectivity map is confirmed in the Sarvian area.

**Figure 5 is about here.**

**Figure 6 is about here.**

**5. CONCLUSION**

The conclusions of this manuscript are as follows.

1) The regression analysis is an appropriate and direct method for MPM by satellite images data. In this paper, the output of processed satellite image using regression analysis indicates the iron potential zones accurately.

2) The application of multivariate regression analysis (as a MPM method) was confirmed in the Sarvian area. This paper used multivariate regression to create a mathematical model (with reasonable accuracy) for iron mineral exploration in the region of interest.

3) Two types of multivariate regression models, as two linear equations, were employed to discover new mineral deposits. According to the results of p-value, $R^2$ and $R^2_{adj}$, the second regression model best fitted observations.

4) The accuracy of the model was confirmed by iron outcrops mapping and geological observations. Based on field observation iron mineralization occurs in contacts between limestone and intrusive rocks (skarn type). Iron mineralization consists dominantly of magnetite, hematite and goethite.

5) The results demonstrate that modelling and mapping satellite images data based on regression analysis and remote sensing data is an efficient approach, as it facilitates timely detection with a low-cost procedure and allows decision makers to decide what necessary action should be taken as the first step in Mineral Prospectivity Mapping (MPM) field.

6) Regression analysis method is a subset of supervised classification due to the mentioned procedure. In this method, target spectrums of training area are used for modeling and MPM.

**ACKNOWLEDGEMENTS**

The authors would like to thank Amirabbas KarbalaeiRamezanali for his helpful suggestions.

**REFRENCES**

Abedi, M., Torabi, S.A., Norouzi, G.-H., Hamzeh,M.: ELECTRE III: a knowledge-driven method for integration of geophysical data with geological and geochemical data in mineral prospectivity mapping. J. Appl. Geophys. 87, 9–18, 2012.

Abedi, M., Torabi, S.A. and Norouzi, G.H.: Application of fuzzy AHP method to integrate geophysical data in a prospect scale, a case study: Seridune copper deposit. Bollettino di Geofisica Teorica, 54, 145–164, 2013.

Abrams, M.: The Advanced Spaceborne Thermal Emission and Reflection Radiometer (ASTER): Data products for the high spatial resolution imager on NASA's Terra platform, International Journal of Remote Sensing, 21, 5, 847-859, 2000.

Allbed, A., Kumar, L., and Sinha, P.: Mapping and Modelling Spatial Variation in Soil Salinity in the Al Hassa Oasis Based on Remote Sensing Indicators and Regression Techniques, Remote Sens.  6, 1137-1157; 2014.

Bonham-Carter, G., Agterberg, F.: Application of amicrocomputer-based geographic information system to mineral potential mapping. Microcomput. Geol. 2, 49–74, 1990.

Carranza, E.J.M.: Geochemical anomaly and mineral prospectivity mapping in GIS, Handbook of Exploration Environmental Geochemistry. Elsevier, Amsterdam, 368 p, 2008.

Carranza, E., Geochemical Anomaly and Mineral Prospectivity Mapping in GIS. Elsevier Science Ltd, Oxford, pp. 351, 2009.

Carranza, E.J.M.: Catchment basin modelling of stream sediment anomalies revisited: incorporation of EDA and fractal analysis. Geochemistry: Exploration, Environment, Analysis, 10, 365–381, 2010a.

Carranza, E.J.M.: Mapping of anomalies in continuous and discrete fields of stream sediment geochemical landscapes. Geochemistry: Exploration, Environment, Analysis, 10, 171–187, 2010b.

Carranza, E.J.M., Sadeghi, M.: Predictive mapping of prospectivity and quantitative estimation of undiscovered VMS deposits in Skellefte district (Sweden). Ore Geol. Rev. 38, 219–241, 2010c.

Feizi, F. and Mansouri, E.: Identification of Alteration Zones with Using ASTER Data in A Part of Qom Province, Central Iran. Journal of Basic and Applied Scientific Research, 2, 73–84, 2012.

Feizi, F. and Mansouri, E.: Separation of Alteration Zones on ASTER Data and Integration with Drainage Geochemical Maps in Soltanieh, Northern Iran. Open Journal of Geology, 3, 134–142, 2013a.

Feizi, F. and Mansouri, E.: Introducing the Iron Potential Zones Using Remote Sensing Studies in South of Qom Province, Iran. Open Journal of Geology, 3, 278–286, 2013b.

Feizi, F., Mansouri, E. and KarbalaeiRamezanali, A.: Prospecting of Au by Remote Sensing and Geochemical Data Processing Using Fractal Modelling in Shishe-Botagh, Area (NW Iran). Journal of the Indian Society of Remote Sensing, 44, 539–552, 2016.

Feizi, F., KarbalaeiRamezanali, A. and Mansouri, E.: Calcic iron skarn prospectivity mapping based on fuzzy AHP method, a case Study in Varan area, Markazi province, Iran. Geosciences Journal, 21, 123–136, 2017.

Ford, A. and Blenkinsop, T.G.: Combining fractal analysis of mineral deposit clustering with weights of evidence to evaluate patterns of mineralization: application to copper deposits of the Mount Isa Inlier, NW Queensland, Australia. Ore Geol. Rev. 33, 435–450, 2008.

Granian, H., Tabatabaei, S. H., Asadi, H. H. and Carranza, E. J. M.: Multivariate regression analysis of lithogeochemical data to model subsurface mineralization: a case study from the Sari Gunay epithermal gold deposit, NW Iran. Journal of Geochemical Exploration, 148, 249–258, 2015.

Golshadi, Z., KarbalaeiRamezanali, A. and Kafaei, K.: Interpretation of magnetic data in the Chenar-e Olya area of Asadabad, Hamedan, Iran, using analytic signal, Euler deconvolution, horizontal gradient and tilt derivative methods. Bollettino di Geofisica Teorica ed Applicata, 57, 329–342, 2016.

Gupta, R. P.: Remote sensing geology, Berlin, Heidelberg: Springer Berlin Heidelberg: Imprint: Springer, 2003.

Li, X., Yuan, F., Zhang, M., Jia, C., Jowitt, S.M., Ord, A., Zheng, T., Hu, X., Li, Y.: Three-dimensional mineral prospectivity modeling for targeting of concealed mineralization within the Zhonggu iron orefield, Ningwu Basin, China, Ore Geology Reviews, doi: 10.1016/j.oregeorev.2015.06.001, 2015.

Lindsay, M.D., Betts, P.G., Ailleres, L.: Data fusion and porphyry copper prospectivity models, southeastern Arizona. Ore Geol. Rev. 61, 120–140, 2014.

Lisitsin, V., González-Álvarez, I., Porwal, A.: Regional prospectivity analysis for hydrothermal-remobilised nickel mineral systems in western Victoria, Australia. Ore Geol. Rev. 52, 100–112, 2013.

Mansouri, E., Feizi, F. and KarbalaeiRamezanali, A.: Identification of magnetic anomalies based on ground magnetic data analysis using multifractal modelling: a case study in Qoja-Kandi, East Azerbaijan Province, Iran. Nonlinear Processes in Geophysics, 22, 579–587, 2015.

Mansouri, E., Feizi, F.: Introducing Au potential areas, using remote  sensing and geochemical data processing using fractal method in Chartagh, western Azarbijan – Iran, E. Archive of  Mining Sciences, Vol., No 2, 397–414, 2016.

Melesse, A. M., Weng, Q., Thenkabail, P. S. and Senay, G. B.: Remote Sensing Sensors and Applications in Environmental Resources Mapping and Modelling, Sensors, 7, 3209-3241, 2007.

Pan, G., Harris, D.P.: Information Synthesis for Mineral Exploration. Oxford University Press, New York, pp.  461 , 2000.

Porwal, A., González-Álvarez, I., Markwitz, V., McCuaig, T., Mamuse, A.: Weights-of-evidence and logistic regression modeling ofmagmatic nickel sulfide prospectivity in the Yilgarn Craton, Western Australia. Ore Geol. Rev. 38, 184–196, 2010.

Xiong, Y., Zuo, R.: Effects of misclassification costs on mapping mineral prospectivity, Ore Geology Reviews, doi: 10.1016/j.oregeorev.2016.11.014, 2017.

Zuo, R., Zhang, Z., Zhang, D., Carranza, E.J.M. and Wang, H.: Evaluation of uncertainty in mineral prospectivity mapping due to missing evidence: a case study with skarn-type Fe deposits in Southwestern Fujian Province, China, Ore Geology Reviews, doi: 10.1016/j.oregeorev.2014.09.024, 2014.

**Table 1.** Wavelength ranges and spatial resolutions of ASTER bands (Abrams, 2000).

| Module | VNIR | SWIR | TIR |
|---|---|---|---|
| | Band 1 0.52 - 0.60 | Band 4 1.650 - 1.700 | Band 10 8.125 - 8.475 |
| | Band 2 0.63 - 0.69 | Band 5 2.145 - 2.185 | Band 11 8.475 - 8.825 |
| | Band 3 N 0.78 - 0.86 | Band 6 2.185 - 2.225 | Band 12 8.925 - 9.275 |
| Spectral bandwidth (μm) | Band 3 B 0.78 - 0.86 (backward looking) | Band 7 2.235 - 2.285 | Band 13 10.25 - 10.95 |
| | | Band 8 2.295 - 2.395 | Band 14 10.95 - 11.65 |
| | | Band 9 2.360 - 2.430 | |
| Spatial resolution (m) | 15 | 30 | 90 |

**Table2.** Formula of regression models used for Aster satellite image bands

| Types of Regression | Number of coefficients | Formula |
|---|---|---|
| First-Degree | 15 | $Y_1 = a_0 + a_1 x_1 + a_2 x_2 + \cdots + a_{14} x_{14}$ |
| First-Degree | 106 | $Y_2 = Y_1 + a_{15} x_1 x_2 + a_{16} x_1 x_3 + \cdots + a_{27} x_1 x_{14} + a_{28} x_2 x_3 + a_{29} x_2 x_4 + \cdots$ $+ a_{39} x_2 x_{14} + a_{40} x_3 x_4 + a_{41} x_3 x_5 + \cdots + a_{50} x_3 x_{14}$ $+ a_{51} x_4 x_5 + \cdots + a_{60} x_4 x_{14} + a_{61} x_5 x_6 + \cdots + a_{69} x_5 x_{14}$ $+ a_{70} x_6 x_7 + \cdots + a_{77} x_6 x_{14} + a_{78} x_7 x_8 + \cdots + a_{84} x_7 x_{14}$ $+ a_{85} x_8 x_9 + \cdots + a_{90} x_8 x_{14} + a_{91} x_9 x_{10} + \cdots + a_{96} x_9 x_{14}$ $+ a_{97} x_{10} x_{11} + \cdots + a_{100} x_{10} x_{14}$ $+ a_{101} x_{11} x_{12} + \cdots + a_{103} x_{11} x_{14} + a_{104} x_{12} x_{13} + a_{105} x_{12} x_{14}$ $+ a_{106} x_{13} x_{14}$ |

**Table 3.** The values for $R^2$ , $R^2_{adj}$ and p-value of ANOVA test of 2 multivariate regression models

| Models | $R^2$ | $R^2_{adj}$ | p-value (ANOVA) |
|---|---|---|---|
| $Y_1$ | 0.738 | 0.715 | 0 |
| $Y_2$ | 0.847 | 0.829 | 0 |

**Table 4.** The calculated coefficients of regression models 1 and 2.

| | Model 1 | | Model 2 | |
|---|---|---|---|---|
| **variables** | Coefficients $(a_i)$ | variables | Coefficients $(a_i)$ | |
| **CST** | 0.275 | CST | 0.677 | |
| $x_1$ | -0.01 | $x_1$ | -0.014 | |
| $x_2$ | -0.12 | $x_2$ | -0.019 | |
| $x_3$ | -0.019 | $x_3$ | -0.045 | |
| $x_4$ | 0.003 | $x_4$ | 0.022 | |
| $x_5$ | -0.006 | $x_5$ | -0.017 | |
| $x_6$ | -0.005 | $x_6$ | -0.001 | |
| $x_7$ | - | $x_7$ | - | |
| $x_8$ | -0.004 | $x_8$ | -0.02 | |
| $x_9$ | -0.005 | $x_9$ | -0.006 | |
| $x_{10}$ | 0.009 | $x_{10}$ | -0.014 | |
| $x_{11}$ | 0.005 | $x_{11}$ | 0.024 | |
| $x_{12}$ | 0.016 | $x_{12}$ | 0.024 | |
| $x_{13}$ | 0.002 | $x_{13}$ | 0.018 | |
| $x_{14}$ | 0.022 | $x_{14}$ | 0.036 | |
| - | - | $x_1 x_4$ | -0.0009 | |
| - | - | $x_1 x_6$ | -0.0002 | |
| - | - | $x_4 x_9$ | -0.0009 | |
| - | - | $x_7 x_8$ | 0.00082 | |

[Figure]

**Fig. 1. a)** The location of the Sarvian area in the Orumieh–Dokhtar magmatic belt, Iran **b)** Geological map of the Sarvian area (scale 1:25000).

[Figure]

**Fig. 2.** Location of the Sarvian iron mine in the study area.

[Figure]

**Fig. 3.** a) Location of training area in the study area. b) Aster satellite image in the training area

(RGB:4,6,8). c) Geological map (scale 1:1000) of training area.

[Figure]

**Fig. 4.** Mineral prospectivity map of the Sarvian area.

[Figure]

**Fig. 5.** Mineral prospectivity map of the Sarvian area which confirmed by iron outcrops.

[Figure]

**Fig. 6.** Mineral prospectivity map of the Sarvian area which confirmed by check field of three target areas.

---

## Referee Comment (RC2) · Anonymous Referee #2 · 11 Aug 2017

In this paper, the authors used multivariate regression analysis as processing tool in ASTER images for enhancing the iron prospectively mapping. In summary, the idea and data look good, the mathematical background appears good. However, there is a considerable lack of motivation, context, focus and impact for the paper. This deficit is compounded by poor organizational structure of all parts especially, the results section. The manuscript is a bit thin in the discussion component of the data analysis and discussion section. I would recommend a major revision of the paper to clearly define the background knowledge, the motivation, the objective and the relevance of the results.

I attached a PDF file for the common revisions.

Details – The abstract has a slight problem with language and writing style such as using of unknown abbreviations (R2 and R2 adj), the studied area and problem definition is missing, clear the importance of the point.

- The introduction lacks motivation and context. There is no clear statement as to the problem at hand (why the work was done) or what is to be learned in terms of geology, iron occurrences in the area, comparison between the applied processing methods and the traditional techniques such as Principle Component Analysis (PCA). Give more details about the importance of the applied technique as previous works. Check the references in the part.

- The study area and geological background, you have write more about the origin and mode of occurrences of the iron zones, what are the common geological structures , trends and their effect in iron zone distributions as shown the geological map?

- In RS Data,. Which spectral bands (channels) of ASTER are used? State the wavelength range. What is the temporal resolution of ASTER images? Which dates of images are used? State the spatial resolution of ASTER image used. Also, you have clear the kinds of image correction which are applied before the processing analysis such as Atmospheric Correction (FLAASH), and mosaicking images, radiometric corrections.

Please also note the supplement to this comment:
https://www.solid-earth-discuss.net/se-2017-25/se-2017-25-RC2-supplement.pdf

**Supplement:**

[revised manuscript text omitted]

---

## Author Comment (AC3) · 15 Aug 2017

Referee Report (RC2) - The abstract has a slight problem with language and writing style such as using of unknown abbreviations (R2 and R2 adj), the studied area and problem definition is missing, clear the importance of the point. Answer: This part corrected.

- The introduction lacks motivation and context. There is no clear statement as to the problem at hand (why the work was done) or what is to be learned in terms of geology,

iron occurrences in the area, comparison between the applied processing methods and the traditional techniques such as Principle Component Analysis (PCA). Give more details about the importance of the applied technique as previous works. Check the references in the part. Answer: This paragraph added to introduction: The aim of this paper is processing of satellite images by mathematical method of regression analyses and using of its applications in remote sensing and geological units. This method was used in the study area due to existence of discovered mineralization (dependent variable) in the region. Therefore, existence of a dependent variable is the main condition of using regression analyses. Due to the fact that Sarvian as a skarn type ore deposit with 8 million tons of magnetite was discovered before, the probability of skarn mineralization in other parts of the study area is also possible. According to the capabilities of regression analyses, Sarvian mine is modeled with the goal of iron exploration in other parts of the study area. First of all, this modeling has been performed with using pixels of mine satellite image as a dependent variable and pixels of other parts as independent variables. Then the relationships between variables has been recognized with the mathematical concepts of regression analyses. Finally, the best fitted model for skarn mineralization in the study area has been recognized for exploration of new iron outcrops.

- The study area and geological background, you have write more about the origin and mode of occurrences of the iron zones, what are the common geological structures, trends and their effect in iron zone distributions as shown the geological map? Answer: This paragraph added to study area: A set of crystallized limestones- dolomites are the oldest geological units with the ages of Permian and Triassic in the study area. Sedimentation of limestone – marl of Qom formation occurred concurrent with continental sedimentation at the Oligocene. Most of tectonic activities in the study area were in the form of vertical movements which causes instability of the basin and changes depth of the sea. Vertical movements at the beginning of Miocene cause volcanic activities which was impressive in the study area. An important magmatism was occurred at the late of Miocene which causes skarn mineralization in the contact of carbonate units

of Qom formation. The main fault of the study area is Bidehend. The Bidehend is a strike-slip fault with a length of 43 kilometers. The Bidehend fault is 10 kilometers away from the study area. The effect of this major fault to the study area is limited to creation of parallel faults and fractures with the same direction of Bidehend fault. There is no relationship between the skarn mineralization and faults in the Sarvian area because no mineralization has been reported in faults and fractures

- In RS Data, Which spectral bands (channels) of ASTER are used? State the wavelength range. What is the temporal resolution of ASTER images? Which dates of images are used? State the spatial resolution of ASTER image used. Also, you have clear the kinds of image correction which are applied before the processing analysis such as Atmospheric Correction (FLAASH), and mosaicking images, radiometric corrections. Answer: The information about the bands, wave lengths and resolution of satellite images are mentioned in 3.1. REMOTE SENSING DATA (INDEPENDENT VARIBLES) and Table1. We added dates of images as you mentioned nicely. In addition the information about image corrections are removed due to dear reviewer comment. - Comments in the attached file: Answer: All comments of SC1 corrected. - In comments part, reviewer asked us to add the unit for Figs 4, 5 and 6. Answer: These images are exports of regression formula. Therefore there isn't any special unit for the results of this formula in export maps. In fact, the values of intervals are depend on similarity rate of dependent pixels to independent pixels. Regards Faranak Feizi